# Mechanistic insights into RNA surveillance by the canonical poly(A) polymerase Pla1 of the MTREC complex

Komal Soni ®[1,4], Anusree Sivadas[2,4], Attila Horvath ®[2], Nikolay Dobrev ®[1], Rippei Hayashi ®[2], Leo Kiss ®[1], Bernd Simon ®[3], Klemens Wild ®[1], Irmgard Sinning ®[1] ✉ & Tamás Fischer ®[2] ✉

The *S. pombe* orthologue of the human PAXT connection, Mtl1-Red1 Core (MTREC), is an eleven-subunit complex that targets cryptic unstable transcripts (CUTs) to the nuclear RNA exosome for degradation. It encompasses the canonical poly(A) polymerase Pla1, responsible for polyadenylation of nascent RNA transcripts as part of the cleavage and polyadenylation factor (CPF/CPSF). In this study we identify and characterise the interaction between Pla1 and the MTREC complex core component Red1 and analyse the functional relevance of this interaction in vivo. Our crystal structure of the Pla1-Red1 complex shows that a 58-residue fragment in Red1 binds to the RNA recognition motif domain of Pla1 and tethers it to the MTREC complex. Structure-based Pla1-Red1 interaction mutations show that Pla1, as part of MTREC complex, hyper-adenylates CUTs for their efficient degradation. Interestingly, the Red1-Pla1 interaction is also required for the efficient assembly of the fission yeast facultative heterochromatic islands. Together, our data suggest a complex interplay between the RNA surveillance and 3'-end processing machineries.

Nuclear RNA surveillance is carried out primarily by the exosome which is involved in 3'−5' RNA degradation and maturation processes[1–3]. While export-competent mature RNAs are packaged into mRNP particles, transcripts that fail to mature properly are rapidly degraded by the nuclear RNA surveillance machinery. In addition, pervasive transcription, occurring widely in eukaryotes largely due to bidirectional promoters, produces tremendous amounts of non-coding RNA transcripts that are degraded by the RNA surveillance machinery[4–7]. These pervasive transcripts, also known as cryptic unstable transcripts (CUTs) in yeast, or promoter upstream transcripts (PROMPTs) and upstream antisense RNAs (uaRNAs) in humans, are barely detectable under steady-state conditions due to their rapid degradation[8,9]. Specific recognition of the wide variety of defective RNAs degraded by the exosome is done by its so-called adaptor complexes[1,10]. The nuclear exosome relies on the helicase activity of the DExH-box RNA helicase Mtr4 to thread single-stranded RNAs into its core, thereby making it a bona fide member of all adaptor complexes[11–16]. The best characterised adaptor complex is the TRAMP complex[17], which was identified as the major exosome targeting factor for CUTs in *Saccharomyces cerevisiae*[18].

Polyadenylation is a post-transcriptional modification of RNA involving the addition of poly(A) tails to the 3'-ends of RNA substrates[19,20]. Following the discovery of the yeast TRAMP complex, the connection between polyadenylation-mediated destabilisation of RNAs as a signal for exosome degradation in eukaryotes was established[21,22]. In humans, two recently identified adaptors known as the nuclear exosome targeting (NEXT)[23] complex and the poly(A) tail exosome targeting (PAXT) connection[24] are responsible for targeting

[1]Heidelberg University Biochemistry Center (BZH), INF 328, D-69120 Heidelberg, Germany. [2]The John Curtin School of Medical Research, The Australian National University, Canberra ACT 2601, Australia. [3]European Molecular Biology Laboratory (EMBL), Meyerhofstr, 1, D-69117 Heidelberg, Germany. [4]These authors contributed equally: Komal Soni, Anusree Sivadas. ✉e-mail: irmi.sinning@bzh.uni-heidelberg.de; tamas.fischer@anu.edu.au

early, unprocessed RNAs, PROMPTs and enhancer RNAs or transcripts with poly(A) tails, respectively.

In *Schizosaccharomyces pombe*, a highly conserved MTREC (Mtl1-Red1 core)[25,26] or NURS (nuclear RNA silencing) complex[27] is responsible for degradation of CUTs and meiotic mRNAs. Notably, the MTREC complex was identified in studies of the Mtr4-like protein 1 (Mtl1) and the zinc-finger protein Red1, which in turn contributed to the definition of the PAXT connection in humans[28]. MTREC is a multi-subunit complex comprising the CAP binding complex Cbc1-Cbc2-Ars2 (CBCA), Iss10-Mmi1, Red5-Pab2-Rmn1 and the canonical poly(A) polymerase Pla1, in addition to the core Mtl1 and Red1 proteins[26]. We have recently described Red1 as the central scaffolding protein of MTREC. Red1 directly interacts with each submodule of the MTREC complex, including Pla1, and connects them to the Mtl1 helicase to deliver the associated RNA cargo to the exosome (ref. [29,30], Supplementary Fig. 1A). Importantly, Pla1 is usually part of the highly conserved cleavage and polyadenylation factor (CPF or CPSF in mammals), where it functions to add poly(A) tails during the 3′-end processing of transcripts[31–33]. Co-purification of Pla1 with the MTREC complex and the extended poly(A) tails in meiotic mRNAs and CUTs that are necessary for their efficient degradation[34–36] therefore appear to be correlated. Consistently, Mmi1, along with Pla1 and poly(A) binding protein Pab2 have been implicated in promoting hyperadenylation of CUTs and meiotic RNAs in fission yeast[35,37]. In addition, Pla1 co-localised and immunoprecipitated with Red1 in an RNA-independent fashion, suggesting a physical interaction between them[36]. Furthermore, it was shown that Red1 and Mmi1 facilitate the hyper-polyadenylation of DSR-containing mRNAs by Pla1[36] and that this hyper-adenylation is required for their efficient degradation.

Studies of Pla1 homologues from mammals (*hs*PAP) and *S. cerevisiae* (*sc*Pap1) show that the polymerase adopts a tri-partite domain architecture[38,39]. It consists of three globular domains, an N-terminal nucleotide binding domain (NTD), a middle domain (MD) and a C-terminal RNA Recognition Motif (RRM)[40], where the NTD and MD form the catalytic centre (Fig. 1A). Detailed kinetic and thermodynamic analyses of *sc*Pap1 suggest that large-scale domain movements in the protein are required for substrate recognition and catalysis[41–43] and the enzyme is stabilised in a closed conformation with extensive contacts between the NTD and RRM domain upon substrate binding[41]. Sequence alignment of Pla1 from different species shows that the NTD and MD are highly conserved between yeast and mammals (66.6% similarity), while the RRM domain is quite diverse (19.5% similarity; Supplementary Fig. 1B). Interestingly, the C-terminal RRM domain lacks the consensus RNA binding residues (reviewed in ref. [44]) and does not interact with the RNA substrate via its canonical RNA binding β-sheet interface, but instead binds via its opposite interface[41]. In *S. cerevisiae*, the RRM domain of *sc*Pap1 is also responsible for binding to the cleavage and polyadenylation subunit *sc*Fip1 (homologous to Iss1 in *S. pombe*)[45,46], tethering it to the CPF. Binding of *sc*Fip1 leads to *sc*Pap1 inhibition in yeast, whereas activation has been observed in mammals and plants[45,47,48].

While Pla1-dependent excessive polyadenylation of meiotic mRNAs that are destined for degradation[34] and hyperadenylation of CUTs has been reported in *S. pombe*[26] (compared to *S. cerevisiae* where MTREC is absent), the functional importance of Pla1 in MTREC has not been studied. In this study, we have used a combination of structural, biochemical and in vivo experiments to understand the specific role of Pla1 as an associated factor of the MTREC complex. Together with nuclear magnetic resonance spectroscopy (NMR) and isothermal titration calorimetry (ITC), we show that the C-terminal RRM domain of Pla1 specifically recognises a 58-residue fragment of Red1. We report crystal structures of Pla1 in its apo form and in complex with Red1. Specific point mutations disrupting the Pla1-Red1 interaction and deletion of the complete Pla1 interacting region in Red1 lead to the widespread accumulation of PROMPTs which also harbour shorter poly(A) tails compared to wild-type cells. Interestingly, the Pla1 "truncated" MTREC complex is also defective in the assembly of facultative heterochromatic islands at meiotic genes. Taken together, our data reveal the molecular details of recruitment of Pla1 by Red1 which leads to hyperadenylation of CUTs, serving as a signal for exosome-mediated target degradation. In addition, in vitro competition experiments between Red1 and Iss1, and additional interactions between MTREC and the CPF-associated factor Msi2 (homologous to CF IB or *sc*Hrp1 in *S. cerevisiae*), indicate the existence of an intricate interaction network between CPF and the MTREC complex to control the processing and surveillance of nascent RNA transcripts.

## Results

### The C-terminal RRM domain of Pla1 mediates its interaction with Red1

We have previously shown that the Red1 fragment comprising residues 240-345 (Red1$_{240-345}$) mediates interaction of Pla1 with the MTREC complex (ref. [29], Supplementary Fig. 1A), but the exact interaction region within Pla1 was not identified. Therefore, we set out to map the minimal interaction region of Pla1 with Red1 using yeast-two hybrid experiments (Y2H). Y2H assays were performed with constructs comprising the NTD and MD of Pla1 (residues 1-352, NTD-MD) or RRM domain of Pla1 (residues 352-566) (Fig. 1B). We found that the Pla1 RRM domain (Pla1$_{RRM}$) is necessary and sufficient for interaction with Red1, consistent with previous reports that the RRM domain of *sc*Pap1 is responsible for mediating protein-protein interaction between *sc*Pap1 and its co-factor, *sc*Fip1[45,46].

To further validate our Y2H results, we decided to reconstitute the Pla1-Red1 complex in vitro. To this end, we used bacterial cell lysates co-expressing His$_6$-MBP fused to various Red1 fragments together with untagged Pla1$_{RRM}$ and assessed the ability of the Red1 constructs to pull down Pla1$_{RRM}$ (Fig. 1C). Red1 is predicted to be rather unstructured in the originally identified Pla1 interacting fragment comprising residues 240-345 (Red1$_{240-345}$) (Supplementary Fig. 1C). Therefore, we used partially overlapping, short fragments of Red1, comprising residues 259-288 (Red1$_{259-288}$), 288-345 (Red1$_{288-345}$) and 288-322 (Red1$_{288-322}$) in addition to Red1$_{240-345}$. While Red1$_{240-345}$, Red1$_{288-345}$ and Red1$_{288-322}$ co-purified Pla1$_{RRM}$, Red1$_{259-288}$ did not, suggesting that residues 288-322 of Red1 contain the Pla1 binding site. We further probed the Pla1-Red1 binding interface using NMR. A two dimensional $^{1}$H, $^{15}$N-HSQC spectrum of Red1$_{288-322}$ shows that this region of Red1 is largely unstructured, as inferred from the low chemical shift dispersion in the $^{1}$H-dimension (Fig. 1D) and analysis of the $^{13}$Cα and $^{13}$Cβ secondary chemical shifts (Supplementary Fig. 2A, B). Subsequently, we performed NMR titrations of unlabelled Pla1$_{RRM}$ into $^{15}$N-labelled Red1$_{288-322}$ and monitored the chemical shifts using two dimensional $^{1}$H, $^{15}$N-HSQC spectra (Fig. 1D). We observed that a majority of the backbone amide resonances of the Red1 peptide show severe line broadening while a handful show chemical shift perturbations (CSPs), both indicative of a clear binding event (Supplementary Fig. S2C, D). To confirm whether Red1$_{288-322}$ contains the complete Pla1 binding site, we performed NMR titrations of unlabelled Pla1$_{RRM}$ into $^{15}$N-labelled Red1$_{288-345}$ (Supplementary Fig. 2E). Surprisingly, we observed additional residues in Red1$_{288-345}$ which show CSPs compared to that of Red1$_{288-322}$. To understand the affinity contribution of these additional residues in Red1$_{288-345}$ and define the exact region of Red1 necessary and sufficient for binding to Pla1, we performed isothermal titration calorimetry. We found that while Red1$_{288-322}$ binds to Pla1 with a $K_D$ = 7.9 μM (Fig. 1E and Supplementary Table 1), the addition of residues 323-345 in Red1$_{288-345}$ leads to a ~5.6-fold increase in binding affinity ($K_D$ = 1.4 μM, Fig. 1F and Supplementary Table 1). Therefore, from the fragments tested in this study, the fragment comprising residues 288-345 binds best to Pla1, and is referred to as the Pla1 interaction region.

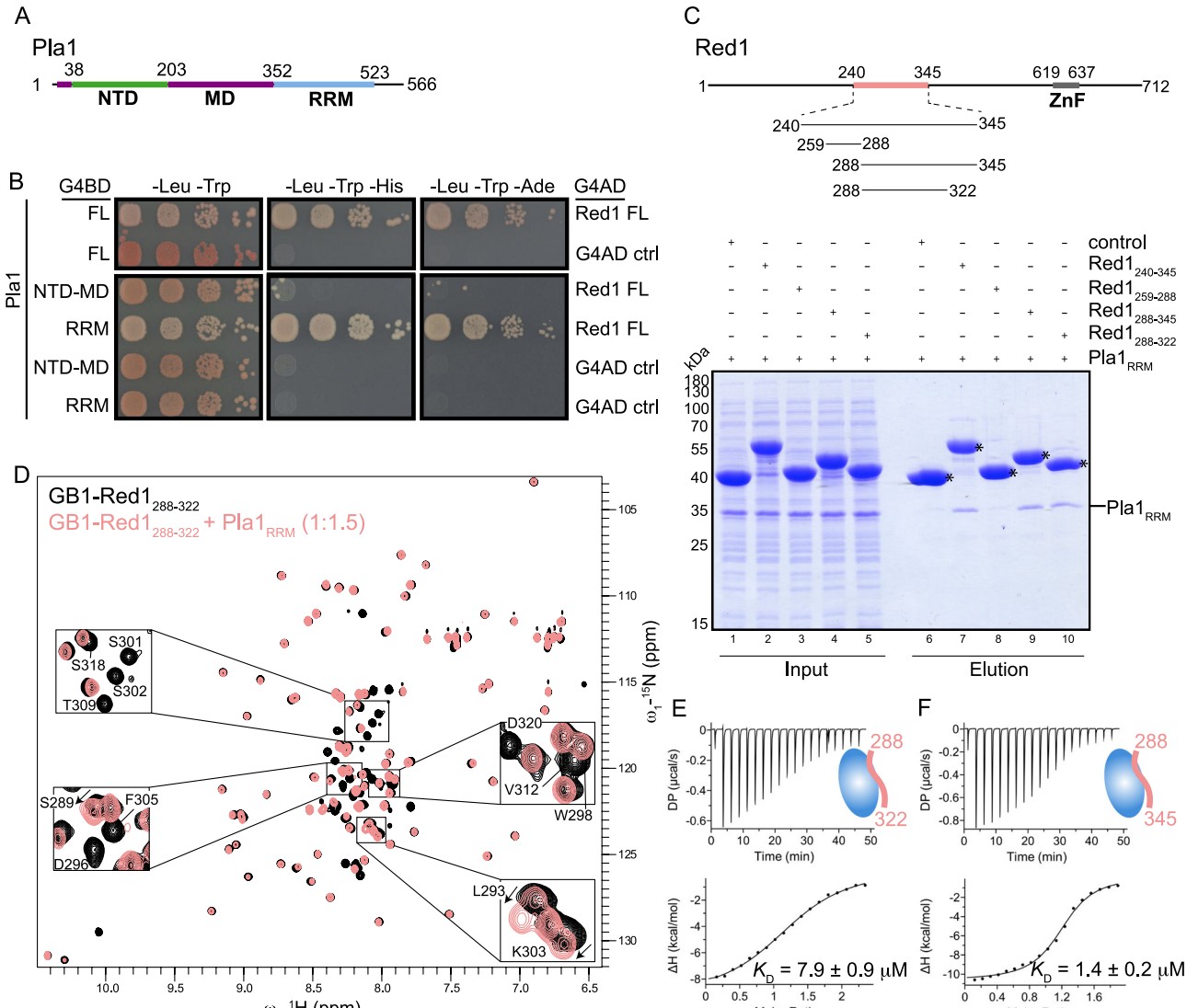

**Fig. 1 | Interaction mapping between Pla1 and Red1. A** Domain organisation of Pla1, consisting of the NTD, MD and RRM domains shown in green, purple and blue, respectively. **B** Y2H experiments show that the Pla1 RRM domain is responsible for the interaction with Red1. Full-length (FL) Pla1, Pla1$_{NTD\text{-}MD}$ or Pla1$_{RRM}$ constructs were fused to the Gal4 DNA binding domain (G4BD) while full-length Red1 was fused to the Gal4 activation domain (G4AD). Auto-activation controls (ctrl) for FL-Pla1 Pla1$_{NTD\text{-}MD}$ and Pla1$_{RRM}$ are provided. Serial dilutions of equivalent amounts of yeast were plated on double (-Leu-Trp) and triple dropout media (-Leu-Trp-His, -Leu-Trp-Ade), with growth on triple dropout media indicating an interaction between the tested proteins. **C** In vitro pull-down assays. Untagged Pla1$_{RRM}$ was co-expressed with His$_6$-MBP fusion constructs of Red1 or His$_6$-MBP (control). 0.01% of input (lanes 1–5) and 25% of elution fractions (lanes 6–10) were separated on 12% SDS-PAGE gel. Asterisks mark the different His$_6$-MBP fusion constructs. A representative gel from two independent runs is shown. Source data are provided as a Source Data file. **D** Overlay of $^1$H, $^{15}$N-HSQC NMR spectra of GB1-tagged Red1$_{288\text{-}322}$ in the absence and presence of Pla1$_{RRM}$ are shown in black and salmon, respectively. Zoom-in views of Red1 residues showing chemical shift perturbations or line broadening are shown. **E, F** ITC experiments with a serial titration of Red1$_{288\text{-}322}$ (**E**) or Red1$_{288\text{-}345}$ (**F**) into Pla1$_{RRM}$. The calculated dissociation constants ($K_D$) from an average of two or three independent measurements are shown.

## Crystal structures of the canonical poly(A) polymerase Pla1 in its apo form and in complex with Red1

To first gain insights into the inter-domain arrangement and interactions between the NTD, MD and RRM domains of Pla1, we determined the crystal structure of the protein in its apo form. We expressed, purified and performed crystallisation trials with the full-length (Pla1$_{FL}$) and a C-terminal truncation (Pla1$_{\Delta14}$) of Pla1. Both constructs readily crystallised under a variety of conditions within 2–4 days. Crystals of Pla1$_{FL}$ diffracted to substantially higher 1.9 Å resolution while Pla1$_{\Delta14}$ diffracted to 2.6 Å resolution (Table 1).

The crystal structure of the full-length protein shows a tripartite domain architecture of Pla1 (Fig. 2A). The overall topology of the three domains is largely similar to that of the Pla1 homologues (Supplementary Fig. 3A). Briefly, the N-terminal catalytic domain is homologous to the catalytic domain of other nucleotidyl-transferases forming a five stranded mixed β-sheet along with four α-helices[49,50]. The middle domain is formed by a four-helix bundle which is capped by the N-terminus of Pla1. The C-terminal RRM domain contains additional secondary structure elements extending the canonical βαββαβ topology of RRMs[44]. It comprises additional helices α14, α15 and β-strands β8, β9 in the extended loop L1 connecting the canonical RRM strands β7 and β10 compared to its homologues (Supplementary Fig. 3B). In our crystal structure, 18 residues of loop L1, loop L2 connecting β10 and α16 and 25 residues at the C-terminus are disordered. Interestingly, residues 415–434, comprising helix α14 and β-strand β9, partially block the canonical RNA binding interface of the RRM domain on one side[40], while the other side is blocked by residues 523–532 belonging to the C-terminus of the protein. The helix α15 of the RRM

**Table 1 | X-ray crystallography data collection and refinement statistics**

| | Pla1_FL | Pla1_Δ14 | Pla1-Red1 |
|---|---|---|---|
| Data collection | | | |
| Beamline | PETRA III, P13 DESY | PETRA III, P13 DESY | PETRA III, P14 DESY |
| Wavelength (Å) | 0.9762 | 0.9762 | 0.9801 |
| Resolution range (Å) | 77.68–1.9 (1.968–1.9) | 57.35–2.599 (2.692–2.599) | 98.02–2.805 (2.905–2.805) (2.955–2.805)[a] |
| Space group | C121 | P1 | P2₁2₁2 |
| Unit cell dimensions | | | |
| a, b, c (Å) | 122.06 66.75 88.12 | 71.79 72.48 73.21 | 128.64 151.34 65.41 |
| α, β, γ (°) | 90 118.18 90 | 99.96 105.70 118.83 | 90 90 90 |
| Total reflections | 333,368 (32,959) | 119,702 (11,522) | 141478 (12,446) 90,056 (3985)[a] |
| Unique reflections | 49,226 (4854) | 34,690 (3409) | 31,789 (654) 20,986 (1049)[a] |
| Multiplicity | 6.8 (6.8) | 3.5 (3.4) | 4.3 (3.8)[a] |
| Completeness (%) | 99.73 (99.02) | 97.04 (95.57) | 63.95 (20.67) 92.4 (96.7)[a] |
| Mean I/sigma(I) | 11.87 (1.53) | 7.26 (0.89) | 9.91 (0.64) 12.8 (2.6)[a] |
| $R_{merge}$ | 0.1145 (1.18) | 0.1545 (1.402) | 0.065 (0.479)[a] |
| $R_{meas}$ | 0.1242 (1.279) | 0.1837 (1.67) | 0.074 (0.555)[a] |
| $R_{pim}$ | 0.04764 (0.4875) | 0.09824 (0.8945) | 0.036 (0.275)[a] |
| $CC_{1/2}$ | 0.998 (0.663) | 0.99 (0.298) | 0.999(0.868)[a] |
| Refinement | | | |
| Reflections used in refinement | 49,223 (4854) | 34,678 (3409) | 20,527 (654) |
| Reflections used for $R_{free}$ | 2462 (243) | 1733 (171) | 1999 (63) |
| $R_{work}$ | 0.1690 (0.3095) | 0.2161 (0.3710) | 0.2413 (0.3401) |
| $R_{free}$ | 0.1923 (0.3453) | 0.2575 (0.4146) | 0.2840 (0.4666) |
| Number of non-hydrogen atoms | 4718 | 8199 | 8531 |
| Macromolecules | 4185 | 8113 | 8528 |
| Ligands | 6 | – | – |
| Solvent | 527 | 86 | 3 |
| R.M.S deviations | | | |
| Bond lengths (Å) | 0.003 | 0.002 | 0.002 |
| Bond angles (°) | 0.56 | 0.51 | 0.49 |
| Ramachandran plot Most favoured (%) | 97.44 | 97.60 | 93.71 |
| Allowed (%) | 2.37 | 2.30 | 5.71 |
| Outliers (%) | 0.20 | 0.10 | 0.57 |
| Rotamer outliers (%) | 0.65 | 1.25 | 1.71 |
| Clashscore | 6.52 | 3.80 | 4.44 |
| Average B-factor | 32.65 | 64.81 | 47.93 |
| Macromolecules | 31.75 | 64.91 | 47.93 |
| Ligands | 36.74 | – | – |
| Solvent | 39.69 | 55.47 | 41.78 |

Statistics for the highest-resolution shell are shown in parentheses.
[a]Statistics reported for ellipsoidal diffraction cut-off.

domain is positioned away from the β-sheet interface facing the back side of the RRM, to form an interface with the Pla1_NTD (Fig. 2A and Supplementary Fig. 4A).

A large interface is formed by a network of interactions between the three domains of the protein burying a total surface area of ~2982 Å². The tripartite domain architecture of Pla1 is held together by hydrogen bonds between the interface residues of the domains (Supplementary Fig. 4A). A comparison of our crystal structures of Pla1_FL and Pla1_Δ14 shows that the individual domains have undergone significant movements. An alignment based on the middle domains illustrates that the N-terminal has a modest rotation of 2.95° with a 1.13 Å displacement, while the C-terminal RRM domain has a more pronounced rotation of 9.8° and 2.26 Å displacement (Supplementary Fig. 4B). Pap1 is also known to undergo large-scale domain motions around the defined hinges connecting the NTD and MD, and the MD and RRM domains[41,43], which are deemed necessary for proper functioning of the enzyme.

To understand the molecular details of the Red1-Pla1 interaction, we determined the crystal structure of the complex (Table 1). Since Red1 binds to Pla1_RRM with low micromolar affinity, we reasoned that the complex should be stably purified using size exclusion chromatography. Indeed, we could reconstitute a highly pure complex which was subjected to co-crystallisation trials (Supplementary Fig. 5A). Co-crystals of the complex appeared within three weeks and diffracted to 2.81 Å resolution. The asymmetric unit contains two molecules of the Pla1-Red1 heterodimer which exhibit small differences between them, as indicated by a root mean square deviation (RMSD) of 0.96 Å. Although the Pla1-Red1 interaction interface is conserved between the two molecules, the overall density for the second heterodimer is much weaker and therefore our structural analyses are based on the first molecule. As in the apo structure, part of loop L1, the complete loop L2 and the C-termini of Pla1_RRM are disordered. RRM domain α15, which forms an interface with the NTD in our apo structure, is also partly disordered. In addition to these, the Red1 C-terminus, spanning residues 323–345, is flexible and therefore not observed in the crystal structure of the complex. A superposition of the apo Pla1_FL and the Pla1-Red1 complex shows that the Pla1_NTD has a rotation of 4.98° with a 1.01 Å displacement, while the Pla1_RRM domain has an angular rotation of 8.06° with a 1.63 Å displacement, again indicative of the flexibility between the domains (Supplementary Fig. 5B).

Our crystal structure of the Pla1-Red1 complex shows that Red1 contacts the C-terminal RRM domain of Pla1 (Fig. 2B). The interaction between Pla1 and Red1 buries a total solvent accessible area of 1338.5 Å². The interaction interface of Red1 can be largely divided into two regions. The first half comprises a short α-helix at the N-terminus (α0) that helps orient Red1 onto Pla1 (Fig. 2C). A network of aromatic stacking interactions occurs between Pla1_RRM Phe468 and Red1 Trp298 and Phe305 (Fig. 2C and Supplementary Fig. 5C). This cluster is further stabilised by hydrophobic stacking of the aliphatic sidechain of Lys368, a hydrogen bond between its side chain and backbone carbonyl of Trp298, a salt bridge formed between Glu416 and Lys294 and packing of Asp513 against Phe305. Importantly, Lys368 positioned close to the C-terminus of Red1 helix α0 reads out the negative dipole of α0 and therefore further stabilises it. The second half of Red1 is largely involved in mainchain-sidechain interactions between Red1 residues Gly306, Ser308, Asn311 and Val312 that pack against Pla1 residues Gln420, Val518, Ile520 and Asn522 (Fig. 2D). In addition, hydrogen bonding interactions occur between Red1 Ser307 and Ser308 side chains with Pla1 Ser424 and Lys506. Interestingly, Red1 Val312-Ile314 form a β-strand (β0) that packs against Pla1 Arg491-Asp493 that also form a short β-strand, leading to the formation of an antiparallel β-zipper[51] (Fig. 2D). Importantly, while our NMR results show that Red1 residues 312–314 have a propensity to form β0 in apo (Supplementary Fig. 2A, B), the Pla1 β-strand is not formed in our Pla1 apo crystal structures. Therefore, this β-strand addition by means of a β-zipper formation stabilises and holds the second half of Red1 in position.

To provide support for the structure of Pla1-Red1 complex and validate its conformation in solution, we recorded SAXS experiments (Supplementary Fig. 5D, Supplementary Table 2). The Kratky plot of the Pla1-Red1 complex shows a bell-shaped curve characteristic of well-

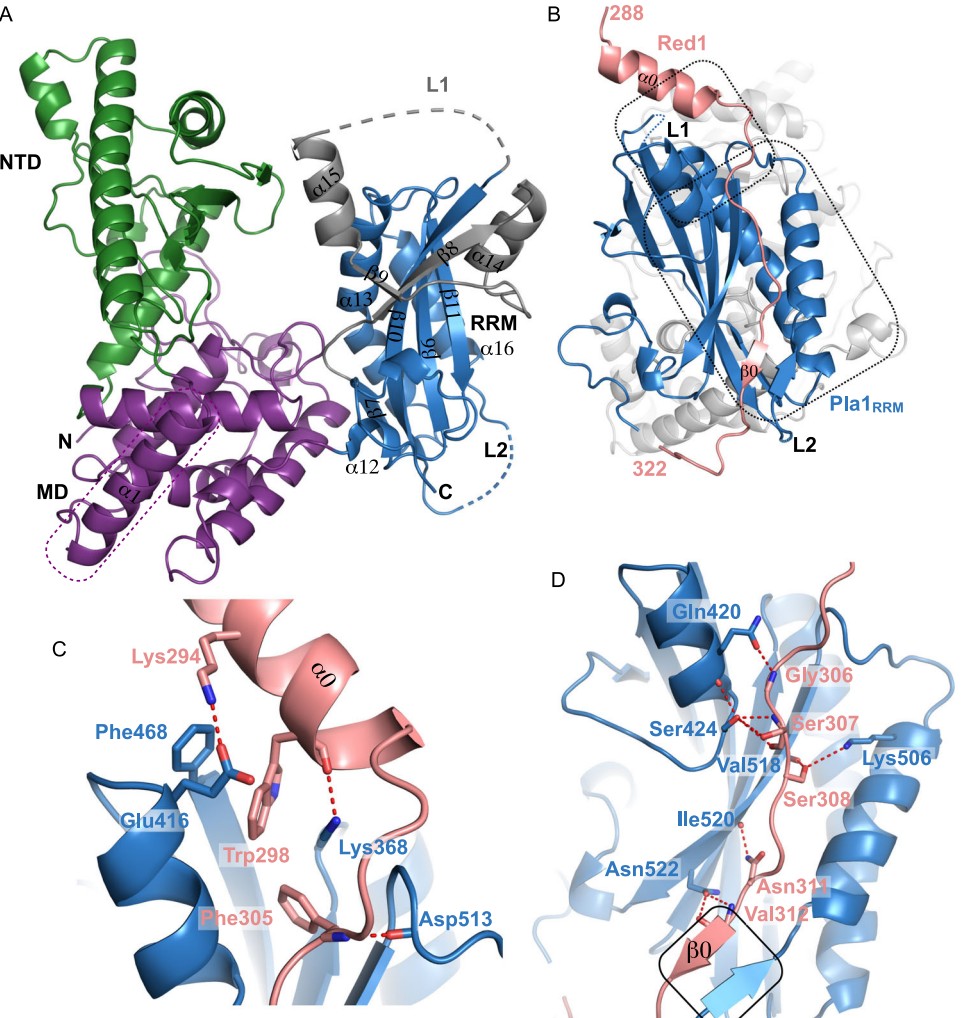

**Fig. 2 | Crystal structures of Pla1 in its apo form and in complex with Red1.**
**A** The overall architecture of Pla1 in its apo form is shown. Colour scheme for Pla1 domains are the same as Fig. 1A. The N- and C-termini of the protein and loops L1 (shown in grey) and L2 of RRM domain are marked. The Pla1 N-terminal α1, which forms a part of the MD, is boxed in purple. The secondary structure elements of the RRM domain comprising helices α12-α16 and β-strands β6-β11 are marked. **B** Crystal structure of Pla1-Red1 complex. Red1 and Pla1$_{RRM}$ are shown in salmon and blue, respectively, while the NTD and MD are coloured in grey. The Red1 N- and C-termini, its secondary structure elements α0 and β0, along with the Pla1$_{RRM}$ domain loops L1 and L2 are marked. The two Red1 interaction interfaces are boxed. **C, D** Zoom-in views of residues involved in the Pla1-Red1 interaction. Polar contacts are shown by red dotted lines. The β-zipper formed between Red1 β0 and Pla1 residues Arg491-Asp493 is boxed in black.

folded globular molecules (Supplementary Fig. 5E). The pairwise distribution curve also shows that the Pla1-Red1 complex has a globular architecture with a $D_{max}$ of 10.7 nm (Supplementary Fig. 5F). Since the Pla1$_{RRM}$ loops L1 and L2, and both Pla1 and Red1 C-termini are missing from the crystal structure, we used CORAL[52] to first model these regions based on the SAXS data (Supplementary Fig. 5G). Subsequent fitting of the experimental SAXS scattering profile of the complex with the crystal structure show that the data are in good agreement with a $\chi^2$ fitting-value of 1.23 (Supplementary Fig. 5H).

**Structure-based mutation analysis in vitro**
Based on the NMR data and our crystal structure of the Pla1-Red1 complex, we performed mutational analyses and evaluated the importance of the affected contacts in ITC measurements using constructs comprising only the Pla1$_{RRM}$ and Red1 residues 288-345 (Supplementary Table 1). We first mutated the aromatic residues in the Red1 N-terminal helix α0 by replacing them with alanine residues to create the double mutant W298/F305A. This double mutant led to a complete disruption of Pla1-Red1 interaction (Fig. 3A). This is not surprising as stacking interactions of Red1 helix α0 residues Trp298

and Phe305 with Pla1$_{RRM}$ Phe468 and Lys368 creates a hydrophobic core which helps orient the Red1 peptide onto Pla1 (Fig. 2C). A charge reversal mutation of Lys368 (K368E) which reads the negative dipole moment of Red1 helix α0 also leads to a ~13-fold loss in affinity (Fig. 3B). Furthermore, mutation of Red1 Val313, which is part of β0-strand, to an arginine residue also leads to ~13-fold loss in affinity (Fig. 3C), signifying the affinity contribution of the β-zipper. In contrast, alanine replacements of Ser308 and Asn311 did not affect the interaction (Supplementary Fig. 6A, B).

Our NMR titrations had indicated that the negatively charged patch at the Red1$_{288-322}$ C-terminus (namely residues Asp317, Ser318, Asp319 and Asp320) also show line broadening upon addition of Pla1$_{RRM}$ (Fig. 1D and Supplementary Fig. 2D). In our crystal structure, however, these residues do not show contacts with Pla1. To probe the importance of these residues for Pla1-Red1 interaction, we made charge reversal mutations of the aspartate residues to arginine (D317/S318/D319/D320 to RSRR, named Red1$_{RSRR}$). To our surprise this triple mutant also completely abrogated Pla1-Red1 binding (Fig. 3D). Since 25 residues at the Pla1 C-terminus are disordered in our crystal structure and the Pla1 and Red1 C-termini are in close proximity to each other

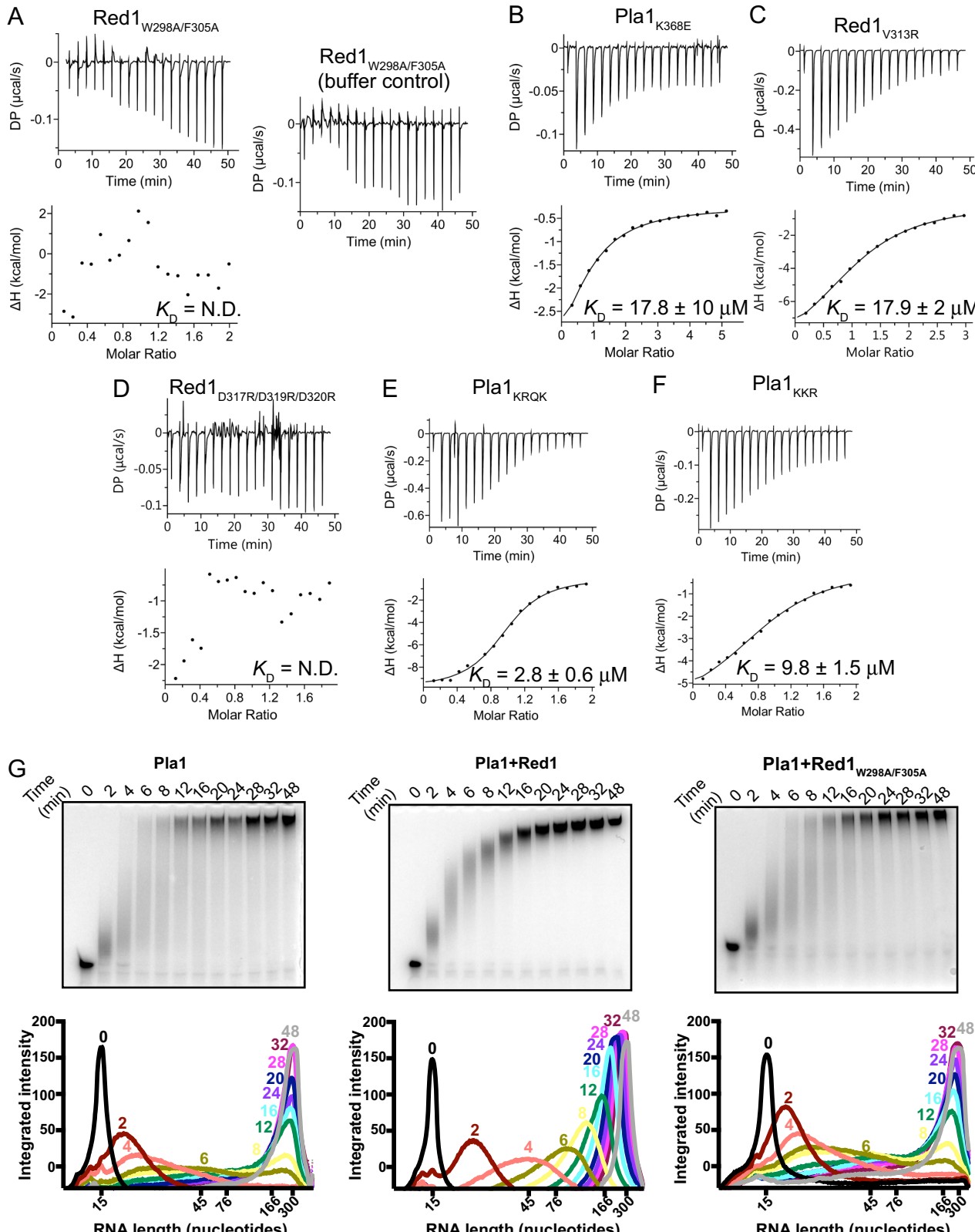

**Fig. 3 | Mutational analyses of Pla1-Red1 interaction.** Isothermal titration calorimetry experiments with specific Pla1 and Red1 point mutants that affect their interaction interface are shown in panels **A**–**F**. The calculated dissociation constant ($K_D$) from an average of two independent measurements is shown. **G** In vitro polyadenylation assay. Polyadenylation of a 5′-Cy3 labelled $A_{15}$ RNA primer by Pla1 alone (left panel), in the presence of $Red1_{288-345}$ (middle panel) and $Red1_{W298A/F305A}$ (right panel), analysed by 14% denaturing urea PAGE at different time points. Densitometric analyses of the gels are plotted below, where the RNA length is marked based on an RNA ladder. Representative gels from two independent experiments are shown. Source data are provided as a Source Data file.

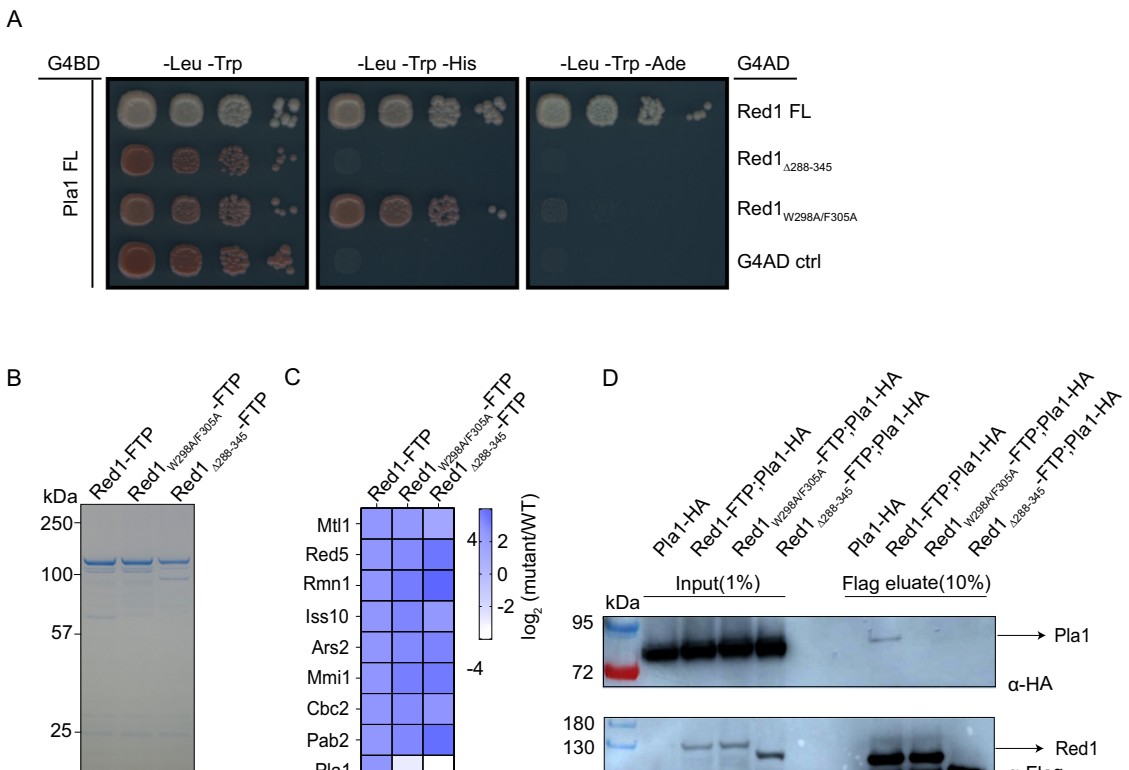

**Fig. 4 | Disrupting Red1-Pla1 interaction in vivo leads to 'Pla1-truncated' MTREC complex. A** Y2H experiments showing the effects of deletion of Red1 residues 288-345 or Red1$_{W298A/F305A}$ double mutant on Pla1-Red1 interaction in the context of full-length proteins. **B** Coomassie blue stained SDS polyacrylamide gel of tandem affinity purified MTREC complex using Red1 as bait, from respective mutant strains (a representative gel from two independent experiments is shown); and **C** heat map representation of mass-spectrometry (MS) analysis of these purifications. Heat map shows the changes in the amount of co-purifying MTREC subunits (except the Cbc1 subunit which was not conclusively quantified), in indicated mutant strains compared to WT. Co-purifying protein amounts were normalised to the corresponding purified bait protein (WT or mutant Red1) amount. **D** Co-

Immunoprecipitation (co-IP) of HA-tagged Pla1 (Pla1-HA) using Red1 as bait in WT and Red1 mutant strains. Inputs and final eluates of the tandem affinity purifications of the indicated strains were blotted to nitrocellulose membrane and cut around the 100 kDa marker. The bottom part of the membrane was probed with α-HA to detect co-purifying Pla1-HA (top panel) and the top part was probed with α-Flag to detect the bait protein Red1-FTP (bottom panel). Note that the Red1 bait protein in the Flag eluate runs ~20 kDa lower than in the input, due to the cleavage of the C-terminal Protein-A tag by the TEV enzyme during the elution step from the IgG column (see details in Methods section). A representative membrane from two independent experiments is shown. Source data are provided in Source Data file.

(Supplementary Fig. 6C), we reasoned that two positively charged patches present in the disordered Pla1 C-terminus could possibly be involved in this interaction. Therefore, we created two Pla1$_{RRM}$ charge reversal mutants: K544/K545/R546 to EEE and K559/R560/Q561/K562 to EEQE, named Pla1$_{EEE}$ and Pla1$_{EEQE}$, respectively. While the Pla1$_{EEQE}$ mutant led to a ~2-fold loss in affinity (Fig. 3E), the Pla1$_{EEE}$ mutant led to a ~7-fold loss in affinity (Fig. 3F), indicating that these charged patches of Pla1 and Red1 C-termini are possibly in close proximity in solution and might be involved in electrostatic interactions. In accordance with these data, structure prediction using ColabFold[53] also shows that Pla1 residues K544/K545/R546 are spatially close to Red1 residues D317/S318/D319/D320 (Supplementary Fig. 6D).

To assess if recombinantly purified Pla1 is active in vitro and if its polyadenylation activity is affected by Red1, we used in vitro polyadenylation assays. Recombinantly purified Pla1 was incubated with 5'-Cy3 labelled A$_{15}$ RNA primer in the absence or presence of Red1 and the reaction was started by addition of ATP. As previously reported, Pla1 is active in in vitro polyadenylation assays[54] (Fig. 3G) compared to a control where the catalytic mutant of Pla1 (D153A) did not show any polyadenylation activity (Supplementary Fig. 6E). Remarkably, Red1 affects the processivity of Pla1 and the poly(A) tail synthesis becomes more distributive, as evidenced by the gradual increase in length of poly(A) of all products compared to a significant difference in the tail lengths of the different RNA species in the absence of Red1. This effect of Red1 on the Pla1 catalytic activity is similar to that observed for the *S.*

*cerevisiae* homologues *sc*Pap1 and *sc*Fip1[45,55]. Of note, the distribution of poly(A) tail lengths of products was similar to wild-type Pla1 when the Pla1-Red1 interaction mutant (Red1$_{W298A/F305A}$) was used (Fig. 3G), indicating that Red1$_{W298A/F305A}$ is unable to bind and therefore influence the processivity of Pla1.

## Disrupting Red1-Pla1 interaction in vivo leads to 'Pla1-truncated' MTREC complex

To assess the effects of Pla1-Red1 interaction mutants, we first performed Y2H experiments using the full-length Red1 containing the W298A/F305A double mutation, together with full-length Pla1 (Fig. 4A). Indeed, these point mutations strongly impaired the Red1-Pla1 interaction, as evidenced by the lack of colonies on -Leu -Trp -Ade media, although yeast colonies on the more sensitive -Leu -Trp -His media indicated a residual binding. However, this interaction was completely abrogated when the Red1 residues 288-345 were deleted (Red1$_{Δ288-345}$; Fig. 4A).

To confirm that the MTREC complex lost its association with its Pla1 subunit in the *red1$_{W298A/F305A}$* and *red1$_{Δ288-345}$* strains in vivo, we replaced the genomic copy of Red1 in *S. pombe* cells with a C-terminally Flag-ProtA-tagged version of wild-type (WT) Red1, *red1$_{W298A/F305A}$* double mutant and *red1$_{Δ288-345}$* deletion mutant. We performed tandem affinity purifications with these strains (Fig. 4B) and analysed the final eluates using mass spectrometry with TMT 10plex mass tag labelling (Fig. 4C). These analyses confirmed that both Red1$_{W298A/F305A}$

and Red1$_{\Delta288\text{-}345}$ proteins maintained their association with MTREC components, purifying similar amounts of MTREC subunits to WT Red1, with the exception of the Pla1 subunit, which was not detectable in the purification of Red1 mutants (Fig. 4C). To further confirm that the Pla1-Red1 interaction was disrupted in the $red1_{W298A/F305A}$ and $red1_{\Delta288\text{-}345}$ mutants, we tagged the genomic copy of Pla1 with an HA-tag in WT and mutant Red1-FTP strain backgrounds and performed co-immunoprecipitation (co-IP) experiments. Western blot (WB) results showed that Pla1-HA signal could not be detected in the final eluate of our Red1-Pla1 interaction mutant strains (Fig. 4D). Taken together, these data confirm that $red1_{W298A/F305A}$ and $red1_{\Delta288\text{-}345}$ mutants lead to a 'truncated MTREC', devoid of Pla1, both in vitro and in vivo.

### Pla1 activity as part of MTREC complex is required for the efficient degradation of PROMPTs

To understand the role of Pla1 in the recognition and degradation of CUTs, we sequenced poly(A)$^{+}$ RNA from WT, Red1 knock-out ($red1\Delta$), $red1_{W298A/F305A}$ and $red1_{\Delta288\text{-}345}$ strains. Metagene plots of sense and antisense RNA levels 500 bp upstream and downstream of all *S. pombe* genes (Supplementary Fig. 7A, B) and a subset of genes (2400 genes, ~40% of all genes) filtered for detectable levels of PROMPTs (Fig. 5A–C) in WT and red1 mutant cells confirmed the strong accumulation of PROMPTs, antisense (AS) RNAs and 3′ intergenic transcripts (3′IGTs) in $red1\Delta$ cells compared to WT. Interestingly, $red1_{W298A/F305A}$ and $red1_{\Delta288\text{-}345}$ cells showed only a moderate but highly reproducible accumulation in the levels of PROMPTs, while AS RNAs and 3′ intergenic transcript levels were not affected. Meiotic genes and levels of intronic sequences were also unaffected in these mutants, compared to WT (Supplementary Fig. 7C–F). To exclude the possibility that changes in the Red1-Pla1 interaction mutants remained undetected due to the complete lack of poly(A) tails in stabilised transcripts, we repeated these analyses, using total RNA sequencing (Supplementary Fig. 8A, B). While the total RNA sequencing detected less antisense RNA transcripts in general, the difference between WT and Red1-Pla1 interaction mutants was nearly identical to the poly(A)$^{+}$ transcriptome analysis, strongly suggesting that the role of Pla1 in the context of the MTREC complex is not the initial poly-adenylation but rather the extension of the poly(A) tails of MTREC substrates, leading to the well documented hyper-adenylation of these transcripts[34–36].

### Disrupting Red1-Pla1 interaction affects the poly(A) tail lengths of CUTs and meiotic mRNAs

To assess poly(A) tail length of CUTs and meiotic mRNAs in WT and Red1-Pla1 interaction mutants, we used Oxford Nanopore Technologies' direct RNA sequencing technology, combined with the *tailfindr* R package[56]. This package can estimate poly(A) tail length of individual reads from the length of the monotonous low-variance raw signal, corresponding to poly(A) tails, at the beginning of each read and combines this information with the unique read ID. After mapping the individual reads to the *S. pombe* genome, we can assign the poly(A) tail length information to these mapped reads, determine mean poly(A) tail length of selected transcripts (e.g., all sequenced transcripts of a particular gene or a particular CUT), and display this information for selected population of transcripts (e.g., CUTs or meiotic mRNAs). Since CUTs and meiotic mRNAs are extremely low abundant in WT cells, and might represent a specific sub-population of these transcripts that escaped MTREC and exosome-mediated degradation, we decided to measure the poly(A) tail length of RNA transcripts associated with the MTREC complex. We have previously shown that RNA immunoprecipitation (RIP) of MTREC complex components strongly enrich CUTs and meiotic mRNAs[26], therefore, we purified Flag-ProtA-tagged WT Red1 as well as Red1$_{W298A/F305A}$ and Red1$_{\Delta288\text{-}345}$ mutants, using conditions that preserve RNA-protein complexes and analysed the co-precipitating RNA transcripts by direct RNA sequencing.

Previous studies evaluated the poly(A) tail length of individual transcripts (individual meiotic mRNAs or individual CUTs[26,34–36]) and concluded that these transcripts are hyper-adenylated, compared to normal mRNAs. Our genome-wide data (Fig. 5D, E) confirm these findings, estimating median poly(A) tail length of mRNAs to be around 31 nucleotides (nts) in WT *S. pombe* cells, while PROMPTs and meiotic mRNAs have nearly twice longer poly(A) tails.

The mean poly(A) tail length distribution of IP-ed mRNA transcripts (likely representing contaminating RNA transcripts in these RIP experiments) shows only a minor difference between WT and mutants (Fig. 5D and Supplementary Fig. 8C; median values are 31, 29 and 27 nts in WT, $red1_{W298A/F305A}$ and $red1_{\Delta288\text{-}345}$ mutants, respectively), while PROMPTs show poly(A) tail length decreased by ~20 nts in $red1_{W298A/F305A}$ and $red1_{\Delta288\text{-}345}$ mutants, compared to WT (Fig. 5E and Supplementary Fig. 8D; median 54 in WT to median 35 and 37 nts in $red1_{W298A/F305A}$ and $red1_{\Delta288\text{-}345}$ mutants, respectively). Interestingly, while Red1-Pla1 interaction mutants did not affect the degradation of AS RNAs or meiotic mRNAs, the median poly(A) tail lengths of these transcripts were also reduced in the mutant strains (Supplementary Fig. 8E–H; meiotic mRNAs: median 56 nts in WT to 32 nts in $red1_{W298A/F305A}$ and 42 nts in $red1_{\Delta288\text{-}345}$ mutants; AS RNAs: median 46 nts in WT to 33 nts in $red1_{W298A/F305A}$ and 40 nts in $red1_{\Delta288\text{-}345}$ mutants). Overall, these experiments show that the poly(A) tail length of CUTs and meiotic mRNAs are reduced by ~20 nts in $red1_{W298A/F305A}$ and $red1_{\Delta288\text{-}345}$ mutants, while the poly(A) tail length of mRNAs remains unaffected.

### Interaction of MTREC complex with CPF

3′-end processing of pre-mRNA involves the complex assembly and action of a number of proteins, including the highly conserved CPF[57–60]. CPF is involved in site-specific endonucleolytic cleavage of the pre-mRNA followed by addition of a poly(A) tail at its 3′ end, which is required for nuclear export of mRNAs. In *S. cerevisiae*, the CPF is composed of three enzymatically active modules of which the polymerase module encompasses, among others, the *sc*Pap1[60] and *sc*Fip1 subunits, and *sc*Fip1 has been suggested to tether *sc*Pap1 to CPF[45,46,61–63]. Consistently, the interaction between *hs*Fip1 and *hs*PAP is also conserved in humans[48]. *sc*Fip1 is an intrinsically disordered protein with short regions directly contacting *sc*Pap1/*sc*Yth1[61–63]. The crystal structure of *sc*Pap1 in complex with a 26 amino acid *sc*Fip1 peptide shows that the N-terminus of the short peptide forms a parallel β-ribbon with *sc*Pap1 and adds an antiparallel β-strand extending the β-sheet interface of *sc*Pap1[61]. Structure alignment of the *sc*Pap1-*sc*Fip1 complex with our *S. pombe* Pla1-Red1 structure shows that *sc*Fip1 and Red1 partially occupy a similar binding interface on the polymerase (Supplementary Fig. 9A). Interestingly, there have been suggestions that the MTREC complex and 3′-end processing machinery act in a coordinated fashion where the CPF complex cleaves the RNA destined for degradation, thereby creating an entry point for the MTREC complex[64,65].

Since the binding interfaces on the polymerase for *sc*Fip1 and Red1 are overlapping, we were wondering if Red1 and Iss1, the fission yeast ortholog of *sc*Fip1, might compete for binding to Pla1. Based on sequence alignment between *sc*Fip1 and Iss1 and the extended *sc*Pap1 binding site recently identified within *sc*Fip1[63] (Supplementary Fig. 9B), we tested the binding of a 47-amino acid construct of Iss1 (residues 30-76) to Pla1 using ITC. We found that Iss1$_{30\text{-}76}$ binds to Pla1 with micromolar affinity ($K_D = 2.5\,\mu M$) (Fig. 6A). In competition experiments, where Red1$_{288\text{-}345}$ is titrated into a pre-formed Pla1$_{RRM}$-Iss1$_{30\text{-}76}$ complex, we observed an ~8-fold decrease in affinity of Pla1 for Red1 ($K_D = 11\,\mu M$; Fig. 6B). In contrast, a titration of Iss1 into Pla1$_{RRM}$-Red1$_{288\text{-}345}$ complex is unable to displace Red1 from the complex (Fig. 6C) compared to a control where the double mutant W298A/F305A of Red1$_{288\text{-}345}$ was used for complex formation (Supplementary Fig. 9C). These data show the existence of negative cooperativity between Iss1 and Red1 due to an

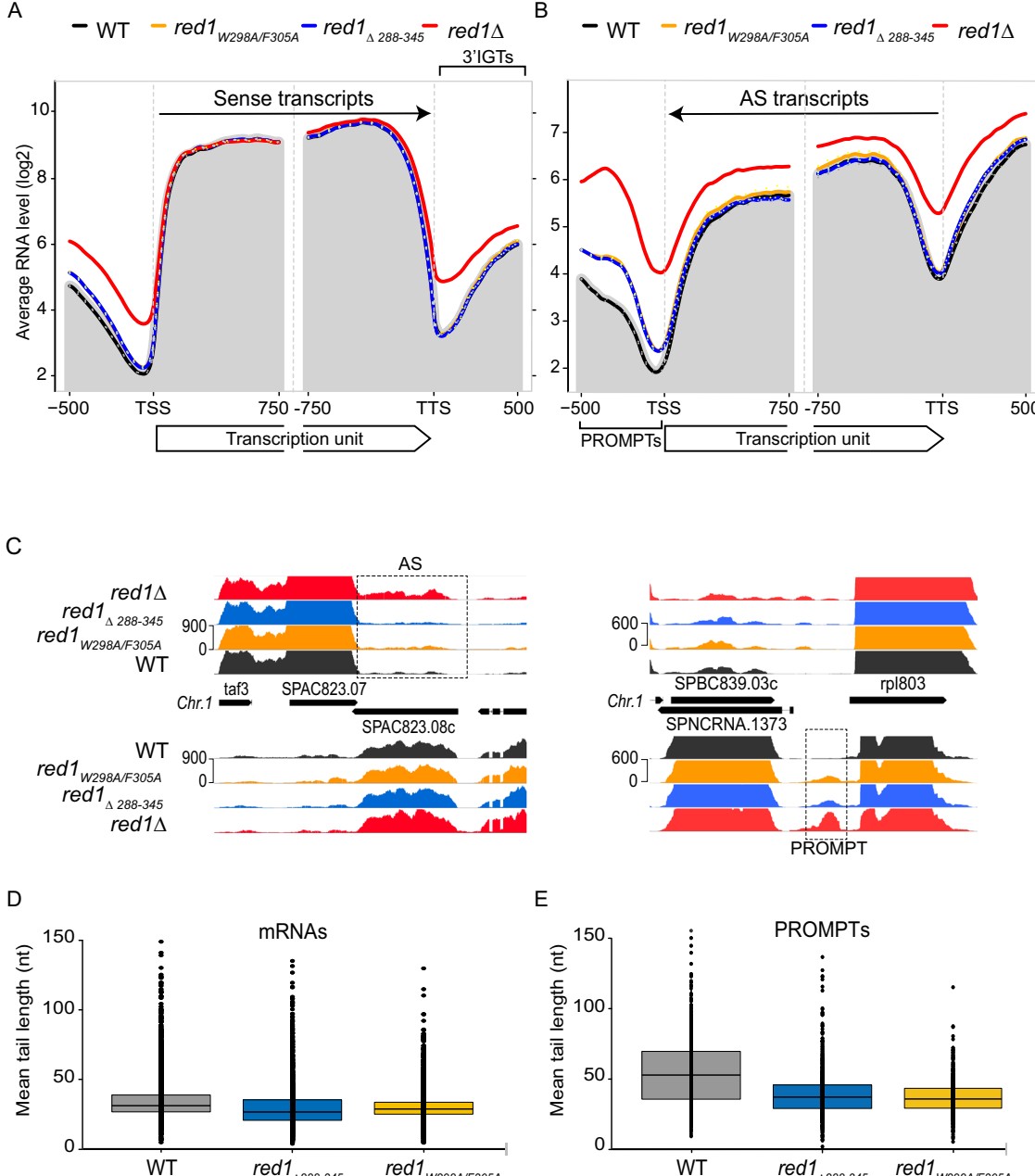

**Fig. 5 | Pla1 in the context of the MTREC complex is responsible for degradation of PROMPTs. A**, **B** Metagene profile of sense (**A**) and antisense (**B**) RNA levels in the indicated strains for a subset of *S. pombe* genes with detectable levels of PROMPTs (2400 genes). The geometric average of RNA levels from 500 bp upstream to 750 bp downstream of the transcription start site (TSS) and 750 bp upstream to 500 bp downstream of the transcription termination site (TTS) are shown. Solid lines represent the average of two replicates for all indicated strains, with the exception of *red1Δ* which represent a single dataset. Dotted lines indicate the individual biological replicates. The grey shading represents the average RNA levels in the WT strain. **C** Strand-specific RNA-seq read coverage of a representative set of genes in WT and mutant strains. Dashed boxes highlight two representative examples of CUTs: antisense RNA transcripts (AS) on the left panel and PROMPTs on the right panel. **D**, **E** Box-plot of mean poly(A) tail length distribution of MTREC-associated mRNAs (**D**) and PROMPTs (**E**) for the WT and indicated mutant strains (5135 mRNAs and 619 PROMPTs with FC > 2 are shown). Dots represent the mean poly(A) tail length of individual mRNAs/PROMPTs, boxes show the 25–75 percentile range and the horizontal lines represent the median values. Upper and lower whiskers represent 75th percentile plus 1.5 times the inter-quartile distance (IQR) and he 25th percentile minus 1.5IQR, respectively. Source data are provided in Source Data file.

overlapping binding interface on Pla1$_{RRM}$. The inability of Iss1 to out-compete Red1 from the Pla1-Red1 complex, even though it binds only -1.8-fold weaker compared to Red1$_{288-345}$, prompted us to investigate the binding kinetics of the two proteins. We therefore used bio-layer interferometry (BLI) to measure possible differences in the $K_{on}/K_{off}$ rates of biotinylated Red1$_{288-345}$ (Fig. 6D) or Iss1$_{30-76}$ (Fig. 6E) immobi-lised on streptavidin-coated biosensors to Pla1$_{FL}$. Compared to Iss1 ($k_{on} = 2.1 \times 10^5 M^{-1}s^{-1}$, $k_{off} = 2.4 \times 10^{-2}s^{-1}$), Red1$_{288-345}$ has a faster

$k_{on}$ ($2.8 \times 10^5 M^{-1}s^{-1}$) and a slower $k_{off}$ ($1.6 \times 10^{-2}s^{-1}$; Supplementary Table 3).

Furthermore, we have previously observed that a member of accessory cleavage factors (CF) of CPF known as Msi2 (*sc*Hrp1 or CF IB in *S. cerevisiae*), co-purifies with MTREC[26], suggesting a possible asso-ciation between these two machineries. We therefore wanted to understand and identify whether Msi2 could directly interact with individual components of MTREC. Indeed, we observed that Msi2

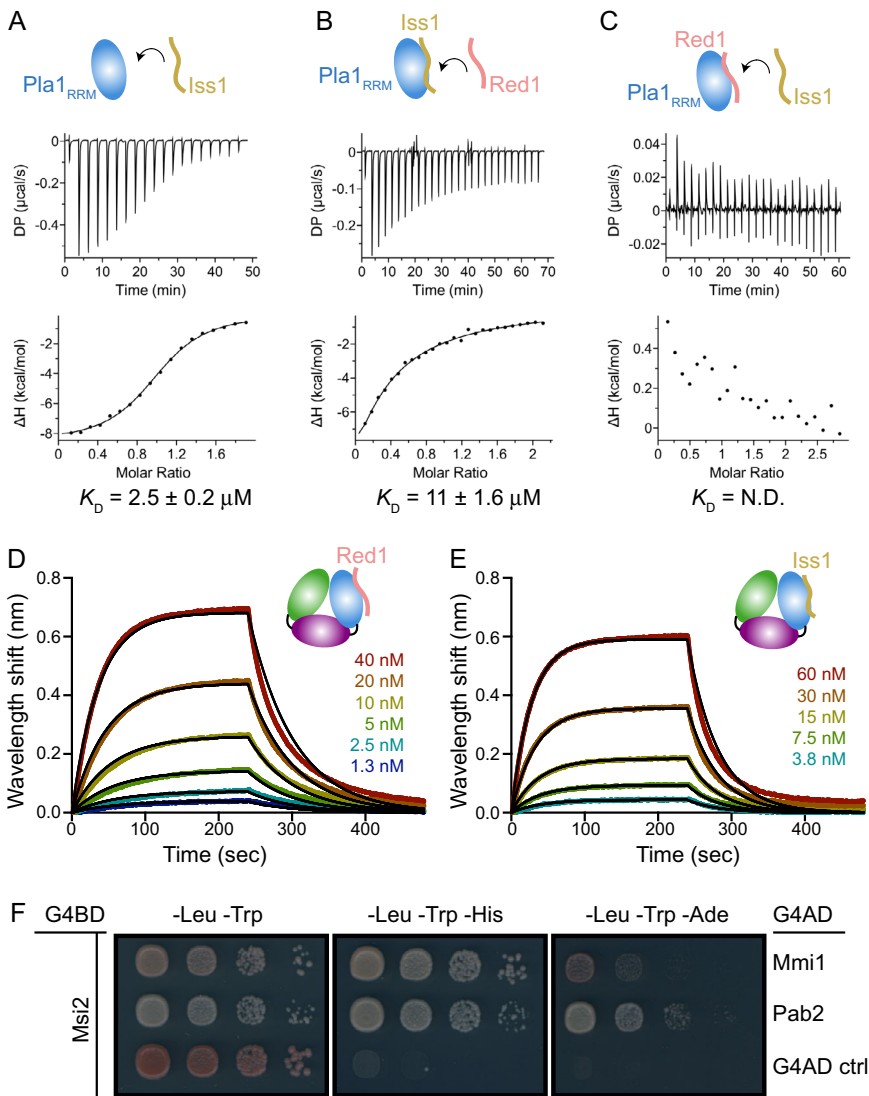

**Fig. 6 | Interactions between MTREC and CPF. A** ITC titration of Iss1 into $Pla1_{RRM}$. **B** ITC titration of $Red1_{288-345}$ into a pre-formed complex of $Pla1_{RRM}$-$Iss1_{30-76}$. **C** ITC titration of Iss1 into a pre-formed complex of $Pla1_{RRM}$-$Red1_{288-345}$. The calculated dissociation constants ($K_D$) from an average of two independent measurements are shown. BLI kinetic analyses of biotinylated $Red1_{288-345}$ and $Iss1_{30-76}$ with varying concentrations of $Pla1_{FL}$ are shown in panels **D** and **E**, respectively. Black lines represent fits to the experimental data obtained using a 1:1 binding global fitting model. **F** Y2H experiments show interaction between Msi2 fused to G4BD and MTREC components Mmi1 or Pab2 fused to G4AD compared to an auto-activation control (ctrl).

binds to two distinct members of the MTREC complex, Pab2 and Mmi1, in Y2H experiments (Fig. 6F and Supplementary Fig. 9D). In summary, these data show that multiple points of association exist between CPF and the MTREC complex and suggest a close functional cooperation between these two complexes in RNA surveillance.

## MTREC-Pla1 interaction is required for heterochromatic island formation at meiotic genes

Recent studies reported the involvement of the CPF, including Pla1, in the formation of small, facultative heterochromatic islands at meiotic genes[64,65]. We wondered if the Red1-Pla1 interaction might play a direct role in this process. We carried out ChIPseq experiments to detect H3K9me2 modifications throughout the fission yeast genome in WT, *red1Δ* and the Red1-Pla1 interaction mutant strains (*red1_{W298A/F305A}* and *red1_{Δ288-345}*). We used 3 independent WT strains in this experiment, with slightly different genetic backgrounds (P1, P419, F3230; see Supplementary Table 4 for genotype information), to account for potential variabilities in the appearance of these facultative heterochromatic islands. We detected all previously described heterochromatic islands

within the *S. pombe* genome[66–69], and their size and H3K9me2 enrichment levels were remarkably uniform between the three independent WT strains (Fig. 7A, B). We also confirmed that most of the facultative heterochromatic islands, mainly located at meiotic genes, are dependent on an intact MTREC complex, as evidenced by the complete absence of these islands in the *red1Δ* cells (Fig. 7B, C and refs. [64,65]). Other islands (non-meiotic islands) are independent of the MTREC complex and they are unaffected in the *red1Δ* strain[66] (Fig. 7D). Interestingly, H3K9me2 enrichment levels at all MTREC-dependent facultative heterochromatic islands were strongly reduced in the Red1-Pla1 interaction mutants, while MTREC-independent islands and the major heterochromatic regions (pericentromeric regions, telomers, mating-type and rDNA-region) were unaffected (Fig. 7B–D). The reduction of the H3K9me2 levels at these islands were comparable to the effect that was reported in CPF mutant strains[64]. This finding is remarkable, since *red1_{W298A/F305A}* and *red1_{Δ288-345}* mutants do not interfere with MTREC-mediated degradation of meiotic mRNAs. These results suggest that while the recruitment of MTREC complex to meiotic mRNAs does not require Pla1 interaction with the MTREC complex, efficient

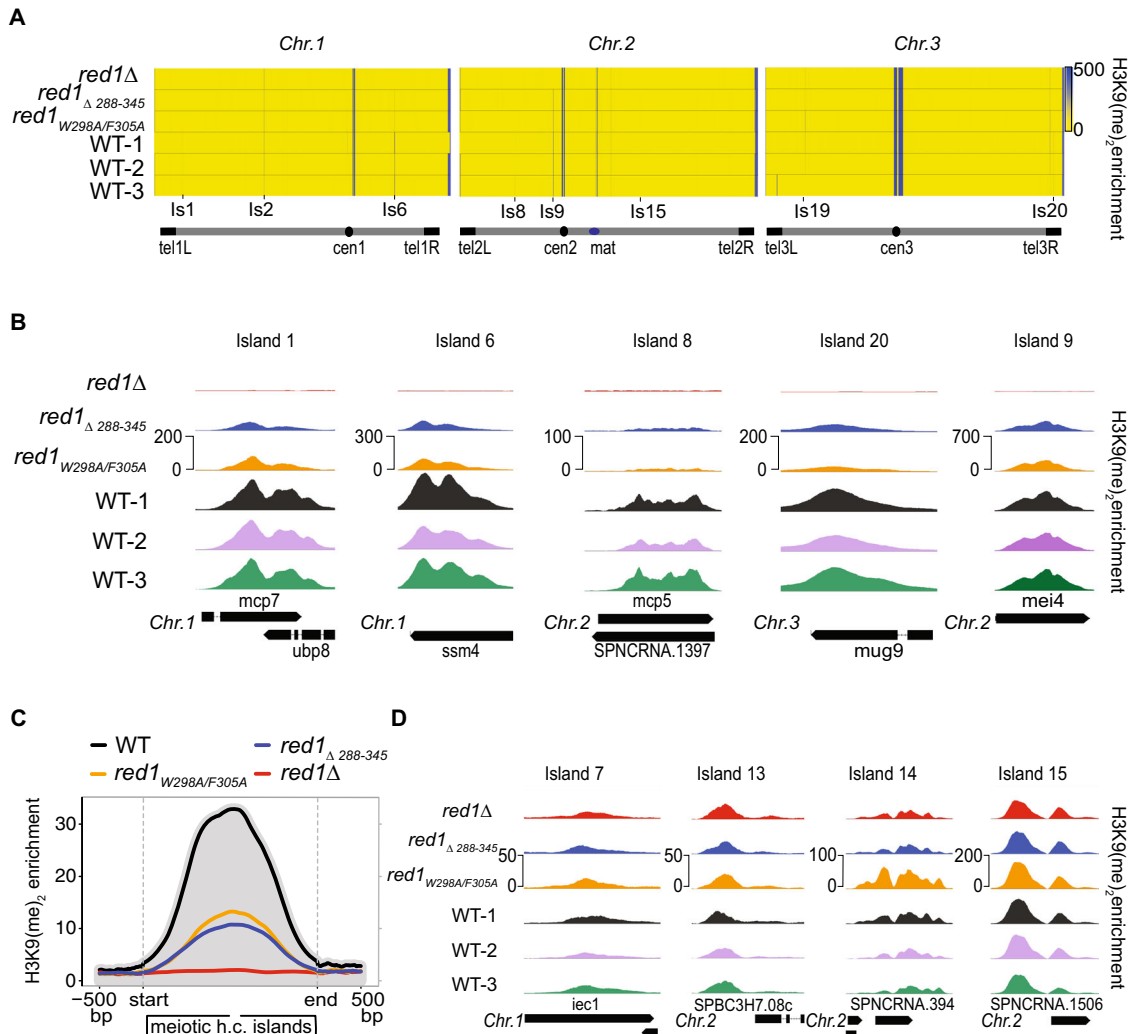

**Fig. 7 | Pla1 – MTREC interaction is required for efficient assembly of facultative heterochromatic islands at meiotic genes. A** Genome-wide heat-map representation of H3K9me2 ChIP-seq analysis of three independent WT strains (WT-1 is the isogenic WT strain for the mutants) and *red1Δ*, *red1_{W298A/F305A}* and *red1_{Δ288-345}* mutants, showing the H3K9me2 enrichment levels (yellow/blue scale of 0-500) for the 3 chromosomes of the *S. pombe* genome (length not to scale). Centromeres, Mating-type locus, telomeres and a selection of facultative heterochromatic islands are indicated. **B** Representative examples of *red1Δ*-sensitive meiotic heterochromatic islands in WTs and mutant strains are shown in higher resolution (scales are indicated for individual islands). **C** Average of H3K9me2 enrichment over all

meiotic heterochromatic islands (12 islands: is1(mcp7); is1.5(tht2); is1.6(SPAC631.02); is2(mug8); is4(SPAC8C9.04/SPNCRNA.925); is5(vps29); is6(ssm4); is8(mcp5); is9(mei4); is16(mbx2/SPNCRNA.1626); is17(mug45); is20(mug9)) plotted on a linear scale. The plots represent the geometric average of enrichment values from 500 bp upstream to 500 bp downstream of the islands, with the island regions scaled to the same lengths for all islands (stretched or condensed to 2000 bp). The grey shading represents the average H3K9me2 enrichment levels in the WT strain. **D** Representative examples of non-meiotic (*red1Δ*-insensitive) heterochromatic islands in WTs and mutant strains (scales are indicated for individual islands).

establishment and/or maintenance of the heterochromatic islands at these loci is dependent on the intact MTREC-Pla1 physical interaction.

## Discussion

The multi-subunit MTREC complex serves as the exosome-adaptor complex required for degradation of CUTs in *S. pombe*. In this study, we identified and characterised the Pla1-Red1 interaction using Y2H, NMR and X-ray crystallography. We showed that the C-terminal RRM domain of Pla1 binds to a 58-residue region of the MTREC core component Red1 (residues 288–345), tethering it to the MTREC complex (Fig. 1). Our crystal structure of the Pla1-Red1 complex showed that Red1 is largely unstructured but comprises an α-helix and a β-strand at the N- and C-termini, which hold Red1 in position to form a tight binding interface with Pla1_{RRM} (Fig. 2B). Using structure-based mutational analyses, we identified a Red1_{W298A/F305A} double mutant in the N-terminal α-helix that severely compromises the Pla1-Red1 interaction in vitro (Fig. 3A) and in vivo (Fig. 4), while deletion of the entire

interaction surface in the Red1_{Δ288-345} mutant leads to no detectable interaction between the MTREC complex and Pla1. Using these Pla1-Red1 interaction mutants, we showed that the truncated MTREC, devoid of Pla1, is unable to hyper-adenylate CUTs and meiotic mRNAs, but interestingly, this only leads to the inefficient degradation of PROMPTs. In addition, the Pla1-Red1 interaction is required for the efficient establishment and/or maintenance of facultative heterochromatic islands around meiotic genes, as evidenced by the strongly impaired levels of H3K9me2 at these loci in the mutants.

Interestingly, similar to the MTREC complex where Pla1 is tethered to the complex via Red1, the *S. cerevisiae* homologue of Pla1 (scPap1) has been shown to be flexibly tethered to the CPF core machinery via the intrinsically disordered protein scFip1[61–63]. Structural superimposition of Pla1-Red1 and the *S. cerevisiae* scPap1-scFip1 complexes show that scFip1 and Red1 occupy a similar binding interface on the respective polymerase (Supplementary Fig. 9A). Surprisingly, scFip1 binds to scPap1 with picomolar affinity[61], while in our

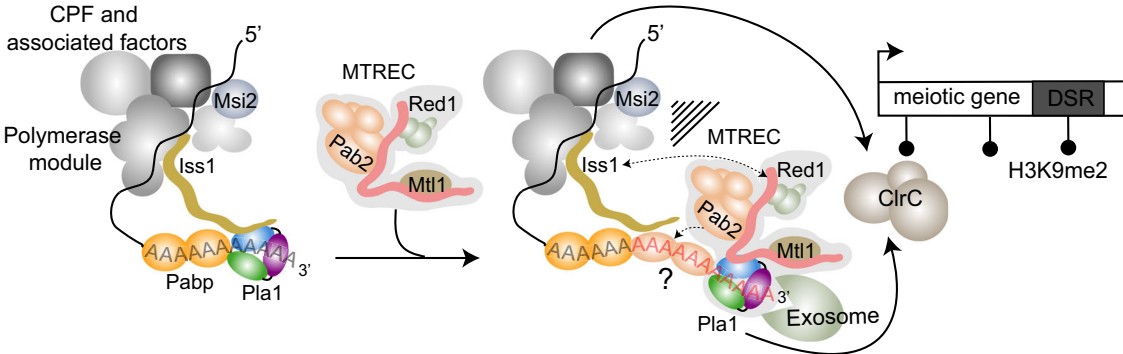

**Fig. 8 | Model of the role of Pla1 in MTREC-mediated degradation of CUTs.** MTREC complex is recruited to CUTs and meiotic mRNAs during their transcription by a not well understood mechanism. Pla1, as part of the CPF, is responsible for the initial poly-adenylation of CUTs and the resulting poly(A) tail is likely bound by the canonical poly(A) binding protein Pabp. MTREC complex sequesters Pla1 from CPF via Red1, replacing Iss1 that anchors Pla1 to CPF. MTREC-bound Pla1 hyper-adenylates CUTs. We suggest that the non-canonical poly(A)-binding protein, Pab2, might be preferentially loaded on these poly(A) tail extensions (marked with a "?" in the Fig.) to facilitate exosome-mediated degradation of CUTs. Both the CPF complex and the Red1-Pla1 interaction are required for the efficient recruitment of the ClrC complex to methylate histone H3K9 at meiotic heterochromatic islands, indicating a sophisticated functional interplay between CPF and MTREC complex during the transcriptional termination, end-processing and degradation of CUTs.

experiments the *S. pombe* homologue of *sc*Fip1 (Iss1) binds with a weaker, low micromolar affinity ($K_D = 2.5\,\mu M$, Fig. 6A), which can be attributed to the low sequence conservation of *sc*Fip1/Iss1 in the polymerase binding region and different assay conditions (ref. [63] and Supplementary Fig. 9B). We find that the binding affinities of Pla1 with Red1 and Iss1 derived using BLI ($K_D = 59.2\,nM$ for Red1, $K_D = 115.2\,nM$ for Iss1) are ~22-24 fold stronger than those obtained using ITC ($K_D = 1.4\,\mu M$ for Red1, $K_D = 2.5\,\mu M$ for Iss1), however the general trend with Red1 binding 2-fold stronger than Iss1 is maintained between the methods. We reason that immobilisation of Red1 or Iss1 on a surface (as done in BLI) decreases the degree of freedom for these proteins leading to an increase in apparent binding affinity as opposed to ITC where the components are mobile and free in solution. In our in vitro competition experiments, we observed that Red1 is able to out-compete Iss1 from its complex with Pla1 (Fig. 6B, C) owing to faster $k_{on}$ and slower $k_{off}$ rates of Red1 when compared to that of Iss1 (Fig. 6D, E and Supplementary Table 3). While these results might differ in the context of the complete CPF and MTREC complexes, there is no evidence to suggest additional interactions between *sc*Pap1/Pla1 and other components of CPF or MTREC, besides *sc*Fip1[63] and Red1[29], respectively. Such a negative cooperation between Iss1 and Red1 is suggestive of Pla1 sequestration from CPF via Red1 to hyper-adenylate CUTs as part of the MTREC complex, although the simultaneous existence of separate copies of Pla1 in these complexes cannot be ruled out (Fig. 8).

Our total RNA sequencing experiments in the Pla1-Red1 interaction mutants did not uncover stabilised transcripts without poly(A) tail, strongly suggesting that the role of Pla1 in the context of the MTREC complex is not the initial poly-adenylation of CUTs, but rather the extension of the poly(A) tail of these transcripts. We carried out poly(A) tail length analyses and showed that the median poly(A) tail length of mRNAs is ~31 nts in WT *S. pombe* cells, while PROMPTs and meiotic mRNAs harbour a poly(A) tail of about 20 nts longer. The median poly(A) tail length of PROMPTs and meiotic mRNAs were 54 and 56 nts, respectively, while AS RNAs showed a somewhat shorter median tail length of 46 nts. It is unclear if PROMPTs and meiotic mRNAs are indeed more extensively hyper-adenylated than other subclasses of CUTs, or whether this lower median value is a data analysis artefact. While PROMPTs and meiotic mRNAs can be annotated and bioinformatically captured relatively easily, intergenic and long antisense CUTs are harder to define and our analysis likely also includes stable ncRNAs which are not hyper-adenylated. A recent study[70] estimated that the median poly(A) tail length of mRNAs is

48.9 nts in *S. pombe* cells. The discrepancy between these numbers (~31 nts in our study versus ~49 nts in ref. [70]) might arise from the fact that we used RNA transcripts isolated from RIP experiments (RNA IPs using WT or mutant Red1-FTP) to enrich low abundant CUTs. Residual RNase activity during the IP procedure might shorten the poly(A) tails in our experimental conditions and we likely underestimate the lengths of the poly(A) tails for all transcripts (both mRNAs and CUTs). Nevertheless, our genome-wide poly(A) tail length analysis confirms that, in WT *S. pombe* cells, CUTs and meiotic mRNAs are hyper-adenylated, similar to previous reports that analysed individual CUTs or meiotic mRNAs[26,34,35,71]. However, in the Pla1-Red1 interaction mutant strains, the poly(A) tail lengths of CUTs and meiotic mRNAs are decreased, while the poly(A) tails of mRNAs (other than meiotic mRNAs) are not affected. In these mutants, all RNA species, including CUTs and mRNAs, have a relatively uniform median poly(A) tail length of ~30 to 35 nts, likely representing the consistent action of Pla1 as part of CPF. These results further support our model that Pla1, as part of the MTREC complex, hyper-adenylates MTREC target RNAs, extending the existing poly(A) tail by ~20 nts (Fig. 8).

It is intriguing why the addition of 20–25 nt poly(A) tails to PROMPTs leads to their efficient degradation, while PROMPTs harbouring shorter poly(A) tails, due to decoupling of Pla1 from the MTREC complex, are inefficiently degraded by the nuclear exosome. Given that the core exosome channel leading to the Dis3 active site can accommodate ~25-30 nts[72], the additional hyper-adenylation of CUTs by MTREC must serve another purpose. One possibility is that hyper-adenylated CUTs are instantaneously bound by the MTREC component poly(A) binding protein 2 (Pab2), making the extended poly(A) tail sterically unavailable for binding to Pabp/Pab1, the major poly(A) binding protein involved in nuclear export and poly(A) tail length control in mRNAs (reviewed in[73]). While Pab2 does not directly influence the polyadenylation activity of Pla1[54], as opposed to its mammalian homologue PABPN1[74,75], it is required for efficient nuclear exosome-mediated degradation of CUTs, meiotic mRNAs and unspliced pre-mRNAs[25,26,71,76,77].

Determination of RNA fate is an extremely complex process requiring timely protein-protein interactions. Our data suggest a close functional interplay between 3'-end processing and the RNA surveillance machinery. However, we found that Msi2 was the only component of the 3'-end processing machinery co-purifying with MTREC complex when we use benzonase treatment, indicating direct protein-protein interactions[26]. Indeed, in our Y2H experiments, Msi2 interacts with two subunits of the MTREC complex, Mmi1 and Pab2 (Fig. 6F).

Importantly in *S. cerevisiae*, *sc*Hrp1 (homologue of Msi2) has been reported to participate in surveillance of CUTs[78], *sc*Nrd1-dependent termination[79] and cytoplasmic nonsense-mediated decay[80], in addition to its canonical role in the 3′-end processing machinery[81]. Whether the interaction between Msi2 and Mmi1/Pab2 helps the recruitment of MTREC to CUTs, or MTREC-bound transcripts are cleaved and poly-adenylated by a specialised form of CPF, remains a topic for future investigations.

In recent years, evidence is emerging for a co-transcriptional interaction between CPF and MTREC components[82]. Both the MTREC complex and multiple subunits of the CPF are required for the estab-lishment and/or maintenance of facultative heterochromatic islands at meiotic genes[27,34,64,65,69,83]. The fact that the Pla1-Red1 interaction mutants strongly destabilise these heterochromatic islands, very similar to the deletion of various CPF subunits, suggests that Pla1, as part of the MTREC complex, must functionally closely interact with CPF, as opposed to having only a strictly downstream function in this process. The exact nature of these interactions and the functional interplay between MTREC, CPF and the ClrC complex that is recruited to these loci to methylate histone H3K9, is not yet understood. Inter-estingly, PAXT component ZFC3H1 (human orthologue of Red1) was shown to functionally interact with the PRC2 complex[84], a key factor in the establishment and maintenance of facultative heterochromatin in higher eukaryotes. *S. pombe* does not have orthologues of PRC com-plexes and H3K27 methylated heterochromatin domains. However, the facultative H3K9me2 heterochromatic islands in *S. pombe* are thought to be the functional predecessor of the PRC-mediated facul-tative heterochromatin domains in higher eukaryotes. Further studies will be required to delineate the mechanistic and functional details of this fascinating interaction between the RNA surveillance machinery and the epigenetic regulation of the genome.

## Methods

### PCR and cloning
The DNA sequence encoding Pla1$_{FL}$ (residues 1-566), Pla1$_{\Delta14}$ (residues 1-542), Pla1$_{D153A}$ (catalytic mutant, residues 1-566) were cloned and ligated into a modified pET24d vector containing non-cleavable N-terminal His$_6$ tag. The C-terminal RRM domain of Pla1 (Pla1$_{RRM}$, resi-dues 352-566) was sub-cloned from Pla1$_{FL}$ into a modified pET24d vector containing N-terminal His$_6$ tag with the Tobacco-Etch-Virus cleavage site present before the corresponding protein. The DNA sequence encoding Iss1 (residues 30-76) was cloned into a modified pET24d vector containing an N-terminal His-Thioredoxin tag upstream of a TEV site. All wild-type and mutant proteins of Red1 with different boundaries were cloned into a modified pET24a vector containing N-terminal GB1 tag-TEV site before the corresponding protein sequence and a C-terminal non-cleavable His$_6$ tag. Site directed mutagenesis with QuickChange Lightning kit was used to introduce point mutations in Pla1 and Red1 proteins, or deletion of residues 288-345 in Red1 and the mutations were confirmed using DNA sequencing.

### Protein expression and purification
The plasmids were transformed in BL21 (DE3) Rosetta2 chemically competent cells for Pla1 and Iss1, grown at 37 °C up to an OD of 1.4–1.6 in Luria broth or Terrific broth subsequently expressed at 20 °C for ~16 hr after induction with 0.5 mM IPTG. For Red1, the plasmids were transformed in BL21 (DE3) chemically competent cells, grown at 37 °C up to an OD of 1 in auto-induction media[85] and subsequently expressed at 20 °C for ~16 h. For isotope-labelled proteins, the bacteria were grown in M9 minimal media supplemented with $^{13}C$-glucose and/or $^{15}NH_4Cl$. Cells were lysed in IMAC buffer containing 20 mM Tris pH 7.5, 200 mM NaCl, 20 mM Imidazole and 2 mM ß-mercaptoethanol. The proteins were purified over 1–2 ml His-Trap FF columns (GE Health-care) with elution buffer containing 250 mM Imidazole. In case of Red1 or Pla1 CTD, overnight tag-cleavage using Tobacco Etch Virus protease

and simultaneous dialysis of the proteins into 20 mM HEPES pH 7.5, 150 mM NaCl, 2 mM BME was carried out. The Red1 proteins were further purified over a second IMAC column where the cleaved pro-teins (still containing C-terminal His$_6$ tag) were eluted using 250 mM Imidazole. For Iss1, purification over a second IMAC column was made to remove uncleaved protein and the fusion tag from the cleaved protein and the flow through was collected. Finally, the Red1 and Iss1 proteins were polished using gel filtration (Superdex 75 16/60) column (GE Healthcare) equilibrated with buffer containing 20 mM HEPES pH 7.5, 150 mM NaCl and 1 mM DTT. For ITC experiments, the DTT was replace by 2 mM BME and for NMR experiments the proteins were purified in buffer containing 20 mM sodium phosphate pH 6.5, 100 mM NaCl, 1 mM DTT. For Pla1 proteins, after the IMAC column the proteins were diluted to buffer containing 50 mM NaCl and further purified using 1 ml Resource Q cation exchange column (GE Health-care), where they were eluted with a linear gradient of 50 mM NaCl to 1 M NaCl. Gel filtration (Superdex 200 16/60) column (GE Healthcare) was used as a final polishing step, where the protein was purified in buffer containing 20 mM HEPES pH 7.5, 150 mM NaCl and 1 mM DTT. For crystallisation of the apo form, the salt concentration was adjusted to 200 mM.

### X-ray crystallography
Pla1$_{FL}$ crystallised at a concentration of 12.7 mg/ml in a drop containing 0.2 M lithium citrate, 20% PEG 3350 at 4 °C as needles within 1 week. Pla1$_{\Delta14}$ crystallised at a concentration of 6.3 mg/ml in a drop containing 0.2 M sodium formate, 20% PEG 3350 at 18 °C as thin plates within 12 days. The Pla1-Red1 complex was prepared by addition of 1.2-fold molar excess of Red1$_{288-345}$ over Pla1$_{FL}$ and subsequent purification over a size exclusion column in buffer containing 20 mM HEPES pH 7.5, 150 mM NaCl and 1 mM DTT, to remove excess Red1. The complex crystallised in a drop containing 0.2 M potassium formate, 20% PEG 3350 at 18 °C as thin plates within 3 weeks. Crystals were flash frozen in mother liquor supplemented with 20% glycerol. Several datasets for the crystals were collected at P13 and P14 beamlines at PETRA III, EMBL Hamburg using mxCuBE v2. Datasets from best diffracting crystals were then processed with XDS software package[86] and the structure was solved by molecular replacement using PDB ID: 2HHP (chain A). The missing residues of Pla1 were built using Coot model building software[87] with multiple rounds of model building and refinement with Phenix[88]. The Pla1$_{FL}$ apo structure was then used as a molecular replacement model for solving the structure of Pla1-Red1 complex. The data were processed with XDS software package[86]. Since the crystal showed anisotropic diffraction (with resolution along the best dif-fracting axis being 2.81 Å and diffraction limits along the other axes being 3.2 Å and 3.8 Å), STARANISO server (http://staraniso.globalphasing.org)[89] was used to apply an anisotropic correction to the data. The anisotropic correction provided an improvement in the overall interpretability of the map especially for the Pla1-Red1 interface residues. Multiple rounds of model building were done in Coot[87] and refinement was performed with Phenix[88].

### NMR spectroscopy
All spectra were recorded at 293 K on Avance III Bruker NMR spec-trometers with proton Larmor frequencies of 600 MHz, 700 MHz or 800 MHz, equipped with cryogenic (600 MHz, 800 MHz) or room temperature (700 MHz) triple resonance gradient probes using Top-spin v3.2. NMR experiments were performed with Red1 constructs (Red1$_{288-345}$ or Red1$_{288-322}$) harbouring a GB1-tag at the N-terminus and a His$_6$-tag at the C-terminus. For backbone assignment of Red1$_{288-322}$, data were recorded on 1 mM protein in buffer containing 25 mM sodium phosphate pH 6.5 and 100 mM NaCl, supplemented with 10% $D_2O$ for lock. Protein backbone resonance assignments were obtained using 3D HNCA, HNCACB, CBCA(CO)NH and HNCO[90]. NMR titrations were done by adding 1.5-fold excess of Pla1$_{RRM}$ into 0.3 mM $^{15}N$-labelled

Red1 constructs (Red1$_{288-345}$ or Red1$_{288-322}$) and recording $^1$H,$^{15}$N HSQC spectra. All experiments were performed at 293 K. Spectra were processed in NMRPipe/Draw[91] and analysed in CCPN Analysis[92].

### Small angle X-ray scattering experiments

The Pla1-Red1 complex was prepared by addition of 1.5-fold molar excess of Red1$_{288-345}$ over Pla1$_{FL}$ and subsequent purification over a size exclusion column in buffer containing 20 mM HEPES pH 7.5, 150 mM NaCl and 1 mM DTT, to remove excess Red1. Peak fractions corresponding to the complex were pooled and concentrated. Measurements were performed at 20 °C at EMBL P12 beamline, PETRA III (DESY, Hamburg, Germany) at concentrations ranging from 0.05-1.7 mg/ml. Forty successive frames with 0.195 s/frame were recorded using an X-ray wavelength of $\lambda = 0.124193$ nm. 1D scattering intensities of samples and buffers were expressed as a function of the modulus of the scattering vector $Q = (4\pi/\lambda)\sin\theta$ with $2\theta$ being the scattering angle and $\lambda$ the X-ray wavelength. Downstream processing after buffer subtraction was done with PRIMUS[93]. $R_g$ was determined using Guinier approximation and from p(r) curve. Disordered regions in the crystal structure were modelled using CORAL[52] and subsequently crystal structure validation was done using CRYSOL[94].

### Isothermal titration calorimetry

ITC experiments were performed with MicroCal PEAQ-ITC calorimeter (Malvern) at 25 °C. Protein samples were dialysed overnight in buffer containing 20 mM HEPES pH 7.5, 150 mM NaCl, 2 mM BME. The cell was filled completely with 40–50 μM protein and the syringe was filled with concentrations in the range of 400–500 μM of the respective ligand. A series of 19 or 26 injections of 2 μl or 1.5 μl titrants were made into the respective protein. For competition experiments, Pla1$_{RRM}$ was pre-incubated with a 1.1-fold molar excess of either Red1$_{288-345}$, W298A/F305A double mutant of Red1$_{288-345}$ or Iss1$_{30-76}$ before loading the sample into the cell. The data were processed with PEAQ-ITC Analysis software (PEAQ-ITC) and were fit to a one-binding site model.

### In vitro polyadenylation assay

A 15-nt polyA chemically synthesised RNA primer with a 5′ Cy3 fluorophore (purchased from iba Life Sciences) was used as a substrate for Pla1. Polyadenylation assays were performed in buffer containing 10 mM HEPES pH 7.5, 20 mM KCl, 5 mM MgCl$_2$, 0.01 mM EDTA, 5% glycerol and 1 mM DTT at 30 °C. 75 nM Pla1 alone or in complex with Red1$_{288-345}$ or W298A/F305A double mutant of Red1$_{288-345}$ at a ~10-fold molar excess was incubated with 200 nM RNA primer for 5 min at 30 °C before starting the reaction with the addition of 2 mM ATP. In all, 10 μl fractions were collected at different time points and mixed with 10 μl denaturing formamide loading dye to stop the reaction. Products were analysed on 14% denaturing urea-PAGE, run in 0.5x Tris-borate-EDTA buffer for 45 min at 35 mA and the gels were imaged using Amersham Imager 600 (GE Healthcare). Densitometric analyses for integration of gel lane intensity were performed using Fiji ImageJ software.

### Yeast two hybrid

The respective sequences of the proteins were cloned into pGBKT7 and pGADT7 vectors from Clontech which harbour either the DNA binding domain or activation domain at the N-termini, respectively. The interaction pairs were analysed by co-transformation into PJ69-4A strain. After a 10-fold serial dilution, colonies were spotted on SDC (SDC-Leu-Trp), SDC-His (SDC-Leu-Trp-His) and SDC-Ade (SDC-Leu-Trp-Ade) plates, incubated at 30 °C and analysed after 3 days. The strength of the interaction was assessed by growth achieved on SDC-His and SDC-Ade as weak and strong, respectively.

### Bio-layer Interferometry

BLI experiments were performed on Octet RED96e system (Fortébio) at 25 °C in buffer containing 20 mM HEPES pH 7.5, 150 mM NaCl, 2 mM BME and 0.01% Tween-20. The protein ligands (Red1$_{288-345}$ and Iss1$_{30-76}$) were biotinylated using EZ-Link NHS-PEG4-Biotin (Thermo Fisher Scientific). Biotin-labelled proteins were immobilised on the streptavidin (SA) biosensors (Fortébio) and a 2-fold serial dilution of Pla1$_{FL}$ was applied to the biosensors. Parallel experiments with a reference sensor where no analyte was added served as control and its signal was subtracted during data analysis. The association and dissociation periods were both set to 240 sec. Data measurements and analysis were performed by using the Data analysis HT 10.0 (Fortébio) software, with a global (full) 1:1 fitting model.

### Tandem affinity purification followed by interaction analysis/ MS

Flag-TEV-protein A (FTP)-tagged bait proteins were harvested from 2 L YEA cultures of yeast strains grown to OD$_{600}$ 1.8–2.2. Cell pellets were snap-frozen in liquid nitrogen and ground into powder using the Cryomill MM-400 (Retsch machine). Cells were resuspended in purification buffer (50 mM HEPES, pH 7.0; 100 mM NaCl; 1.5 mM MgCl2; 0.15% NP-40), supplemented with 1 mM dithiothreitol (Sigma), 1 mM phenylmethylsulphonyl fluoride (Sigma Aldrich, 78830) and protease inhibitor mix (SERVA, 39104). Cell extracts were centrifuged at 3500 g for 10 min at 4 °C, then the supernatants were further centrifuged at 27,000×$g$ for 45 min at 4 °C. Clarified supernatants were then incubated with 150 L slurry of IgG beads (GE Healthcare, 17-0969-02) for 2 h at 4 °C on a turning wheel. After binding, the beads were washed with 2 × 15 mL purification buffer and TEV cleavage was performed in purification buffer containing 20 units AcTEV protease (Invitrogen 12575015), 0.5 mM dithiothreitol and 50 units Benzonase (Sigma, E1014) for the removal of nucleic acids, for 2 h at 16 °C. The eluate was collected and incubated with 100 μL slurry of anti-Flag beads (Sigma Aldrich, A2220) for 1 h at 4 °C. The protein- bound anti-Flag beads were washed with 2 × 10 mL purification buffer and the proteins were subsequently eluted from the beads by competition with 200 μL Flag peptide (Assay Matrix, A6001). The resulting eluate was collected and used for total protein isolation using Trichloroacetic acid- TCA (Sigma, T0699). The eluted proteins were then analysed by Coomassie staining using Brilliant Blue G colloidal concentrate (Sigma Aldrich, B2025) on 4–12% NuPAGE Bis-Tris gels (Life Technologies).

For interaction analysis, the protein samples separated by SDS-PAGE gel was transferred onto a nitrocellulose (Invitrogen, IB301002) membrane, the membrane was cut around 100 kDa marker and the two parts were probed with HRP anti-HA antibody (Abcam ab1190) or HRP anti-Flag M2 antibody (Sigma-Aldrich A8592) in 1:2000 dilution. For mass spectrometry analysis, the precipitated protein samples were subjected to an in-solution tryptic digest using a modified version of the Single-Pot Solid-Phase-enhanced Sample Preparation (SP3) protocol[95,96]. In total three biological replicates were prepared including control, wild-type and mutant derived lysates (n = 3). Lysates were added to Sera-Mag Beads (Thermo Scientific, #4515-2105-050250, 6515-2105-050250) in 10 μl 15% formic acid and 30 μl of ethanol. Binding of proteins was achieved by shaking for 15 min at room temperature (RT). SDS was removed by 4 subsequent washes with 200 μl of 70% ethanol. Proteins were digested overnight at room temperature with 0.4 μg of sequencing grade modified trypsin (Promega, #V5111) in 40 μl Hepes/NaOH, pH 8.4 in the presence of 1.25 mM TCEP and 5 mM chloroacetamide (Sigma-Aldrich, #C0267). Beads were separated, washed with 10 μl of an aqueous solution of 2% DMSO and the combined eluates were dried down. Peptides were reconstituted in 10 μl of H$_2$O and reacted for 1 h at room temperature with 80 μg of TMT10plex (Thermo Scientific, #90111)[97] label reagent dissolved in 4 μl of acetonitrile. Excess TMT reagent was quenched by the addition of 4 μl of an aqueous 5% hydroxylamine solution (Sigma, 438227). Peptides were reconstituted in 0.1% formic acid, mixed to achieve a 1:1 ratio across all TMT-channels and purified by a reverse phase clean-up step (OASIS HLB 96-well μElution Plate, Waters #186001828BA).

Peptides were subjected to an off-line fractionation under high pH conditions[96]. The resulting 12 fractions were then analysed by LC-MS/MS on an Orbitrap Fusion Lumos mass spectrometer (Thermo Scientific) as previously described[98]. To this end, peptides were separated using an Ultimate 3000 nano RSLC system (Dionex) equipped with a trapping cartridge (Precolumn C18 PepMap100, 5 mm, 300 µm i.d., 5 µm, 100 Å) and an analytical column (Acclaim PepMap 100. In all, 75 × 50 cm C18, 3 mm, 100 Å) connected to a nanospray-Flex ion source. The peptides were loaded onto the trap column at 30 µl per min using solvent A (0.1% formic acid) and eluted using a gradient from 2 to 40% Solvent B (0.1% formic acid in acetonitrile) over 2 h at 0.3 µl per min (all solvents were of LC-MS grade). The Orbitrap Fusion Lumos was operated in positive ion mode with a spray voltage of 2.4 kV and capillary temperature of 275 °C. Full scan MS spectra with a mass range of 375–1500 m/z were acquired in profile mode using a resolution of 120,000 (maximum fill time of 50 ms or a maximum of 4e5 ions (AGC) and a RF lens setting of 30%. Fragmentation was triggered for 3 s cycle time for peptide like features with charge states of 2–7 on the MS scan (data-dependent acquisition). Precursors were isolated using the quadrupole with a window of 0.7 m/z and fragmented with a normalised collision energy of 38. Fragment mass spectra were acquired in profile mode and a resolution of 30,000 in profile mode. Maximum fill time was set to 64 ms or an AGC target of 1e5 ions). The dynamic exclusion was set to 45 s.

Acquired data were analysed using IsobarQuant[99] and Mascot V2.4 (Matrix Science) using a reverse UniProt FASTA Schizosaccharomyces pombe database (UP000002485) including common contaminants. The following modifications were taken into account: Carbamidomethyl (C, fixed), TMT10plex (K, fixed), Acetyl (N-term, variable), Oxidation (M, variable) and TMT10plex (N-term, variable). The mass error tolerance for full scan MS spectra was set to 10 ppm and for MS/MS spectra to 0.02 Da. A maximum of 2 missed cleavages were allowed. A minimum of 2 unique peptides with a peptide length of at least seven amino acids and a false discovery rate below 0.01 were required on the peptide and protein level[100], resulting in 164 proteins. Raw TMT reporter ion intensities (signal_sum columns) were first cleaned for batch effects using limma[101] and further normalised using vsn (variance stabilization normalisation[102]. Proteins were tested for differential expression using the limma package. The replicate information was not added as a factor in the design matrix since some condition were measured with a single replicate only. A protein was annotated as a hit with a false discovery rate (fdr) smaller 5% and a fold-change of at least 100% and as a candidate with a fdr below 20% and a fold-change of at least 50%.

## Total RNA isolation

Total RNAs were isolated using TriReagent (Sigma-Aldrich, T9424). Briefly, cell pellets measuring an $OD_{600}$ of 8 was resuspended in Tri Reagent and lysed, treated twice using 1-Bromo-3-chloropropane (Sigma-Aldrich, B9673) followed by RNA precipitation using 2-propanol (Sigma-Aldrich). The RNA pellet was washed in ice-cold 75% ethanol and solubilized in nuclease-free water. The RNA concentration was measured using NanoDrop (Thermo Scientific) and 10 µg of RNA was used for DNase treatment using NEB DNase 1 (M0303). The DNase treated samples were purified further using the RNA Clean and Concentrator Kit (Zymo Research, R1015/R1017) and stored at −80 °C after addition of ribonuclease inhibitor (Invitrogen, 10777019).

## Illumina sequencing of poly(A) + RNA

Total RNA was isolated, DNase treated and purified from respective strains as mentioned above. The RNA quality was assessed using Bioanalyzer (Agilent 2100) and only those of high quality was proceeded to library preparation. Briefly, 1 µg was taken from individual samples, and ERCC-RNA spikeIn (Life technologies, 4456740) was added according to manufacturer's instructions. The RNA samples were then subjected to OligoT purification of poly(A) RNA (NEB, E7490) following the manufacturer's instructions and the recovered poly(A) selected RNA was used for cDNA library preparation using NEBNext Ultra II directional RNA library kit (NEB, E7760) following library preparation protocol. The final cDNA libraries were amplified (cycle number of 8) using NEBNext Multiplex Oligos for Illumina (NEB, E7335) and purified using SPRIselect size selection beads (Beckman Coulter, B23317). Individual library quality was assessed on TapeStation (Agilent 4200) using a DNA D1000 High sensitivity tape. Indexed libraries were pooled and sequenced with ~20 M reads per sample on a NovaSeq. sequencer with 150 bp paired-end reads.

## Illumina sequencing of total RNA

Total RNA was isolated, DNase treated and purified from respective strains as mentioned above. The RNA quality was assessed using Bioanalyzer (Agilent 2100) and only those of high quality was proceeded to library preparation. Briefly, 100 ng was taken from individual samples, and an ERCC-RNA spikeIn (Life technologies, 4456740) amount equivalent to that for 1 µg of RNA input (equivalent to that of poly(A) + sample) was added. The RNA samples were directly proceeded for cDNA library preparation using NEBNext Ultra II directional RNA library kit (NEB, E7760) following the library preparation protocol. The final cDNA libraries were amplified (cycle number of 6) using NEBNext Multiplex Oligos for Illumina (NEB, E7335) and purified using SPRIselect size selection beads (Beckman Coulter, B23317). Individual library quality was assessed on TapeStation (Agilent 4200) using a DNA D1000 High sensitivity tape. Indexed libraries were pooled and subjected for deep sequencing with ~80 M reads per sample on a NovaSeq. sequencer with 150 bp paired-end reads.

## Nanopore sequencing of poly(A) + RNA

Total RNA was isolated, DNase treated and purified from respective strains as mentioned above. The RNA quality was assessed using Bioanalyzer (Agilent 2100) and only those of high quality was proceeded to library preparation. Briefly, 5 µg of RNA samples was subjected to OligodT purification of poly(A) RNA (NEB, E7490) following the manufacturer's instructions and 100 ng of recovered poly(A) + selected RNA was used for library preparation using Oxford Nanopore direct RNA sequencing library kit (SQK-RNA002; Version: DRS_9080_v2_revM_14Ag2019) protocol. The prepared libraries were run on individual flow cells following manufacturer guidelines.

## RNA- Immunoprecipitation followed by sequencing (Illumina and Nanopore)

Tandem affinity purifications of FTP-tagged strains were performed as described above with minor modifications. Reagents were prepared under RNase-free conditions, in the presence of RNase inhibitor (Invitrogen, 10777019). Two-third of final flag elute was taken for DNase treatment using NEB DNase 1 (M0303) and RNA was purified using RNA Clean and Concentrator Kit (Zymo Research, R1015/R1017). The final RNA concentration was determined using Qubit RNA high Sensitivity Assay Kit and 100 ng of RIP-purified RNA was used for cDNA library preparation using NEBNext Ultra II directional RNA library kit (NEB, E7760) following library preparation protocol. The final cDNA libraries were amplified using NEBNext Multiplex Oligos for Illumina (NEB, E7335) and purified using SPRIselect size selection beads (Beckman Coulter, B23317). Individual library quality was assessed on TapeStation (Agilent 4200) using a DNA D1000 High sensitivity tape. Indexed libraries were pooled and sequenced on NovaSeq sequencer with 150 bp paired-end reads. Similarly, 200 ng of RIP-purified RNA was used for library preparation using Oxford Nanopore direct RNA sequencing library kit (SQK-RNA002; Version: DRS_9080_v2_revM_14Ag2019) protocol. The prepared libraries were run on individual flow cells following manufacturer guidelines.

## RNA-seq analysis

Paired-end illumina reads were aligned with hisat 2.1.0[103] allowing introns with a maximum length of 2000 nt ('−max-intronlen 2000'). Aligned bam files were then sorted and indexed using samtools 1.3.1[104] Nanopore reads were aligned with minimap (2.10) with long-read spiced alignment (with splice and -k7 parameters). Strand-specific bigwig tracks were generated using bamCoverage[105] to the pombe genome (ASM294v2) with the parameter '−binSize 1'. Bigwig tracks were normalised by the sum of the raw signals of chromosome I and II multiplied by 100 million. Meta plots were generated with in-house R scripts using GenomicRanges[106] packages. Integrative Genomics Viewer (IGV2.3, Broad Institute) was used for data browsing and creating representative snapshots.

## Detection and quantification of CUTs and PROMPTs

Putative CUT regions were created by neighbouring signal islands closer than 25nts and showing FC > 1.2 relative to red1Δ (length ≥100 nts) using Rsamtools[107] and dplyr[108] separately on the forward and reverse bigwig tracks. PROMPT regions were defined as CUTs that are shorter than 1500 nts and in the + −250nts vicinity of an annotated pombe gene (ASM294v2) on the opposite strand. Intersections, subtractions and merging of the predicted regions were done with Bed-Tools 2.28.0.

## Quantification of polyA length from poly(A) + RNA Nanopore sequencing

polyA length for the whole transcriptome, meiotic genes, CUT and PROMPT regions was estimated using tailfindr[56] and boxplots of arithmetic mean of signals per regions were plotted with the ggplot2 R package.

## Chromatin Immunoprecipitation-sequencing

*S. pombe* strains grown to an $OD_{600}$ of 0.5−0.8 were crosslinked using formaldehyde solution (Sigma, F1635) to a final concentration of 1% at RT for 15 min and quenched by addition of glycine (ThermoFisher, AJA1083) to a final concentration of 150 mM for 5 min at RT. The crosslinked culture was then washed and harvested twice using 1x PBS (137 mM NaCl, 2.7 mM KCl, 10 mM $Na_2HPO_4$, 1.2 mM $KH_2PO_4$) by centrifugation at 3500 g for 2 min at 4 °C each and the final cell pellet was frozen in liquid nitrogen. The cell pellet was then resuspended in 300 µL FA-SDS buffer (50 mM HEPES-KOH pH 8, 2 mM EDTA pH 8, 150 mM NaCl, 1% TritonX-100, 0.1% Sodium Deoxycholate, 0.1% SDS) supplemented with 1x protease inhibitor mix (SERVA, 39104), 1 mM phenylmethylsulfonyl fluoride (Sigma Aldrich, 78830) was homogenised at 4 °C using 700 µL zirconia beads (BioSpec, 110791) in Precellys 24 homogenizer (Bertin Technologies, France) at 5500 rpm. The lysate was collected and transferred into the Covaris AFA Fiber&cap 12 × 12 mm millitube (Covaris, 520135) after bringing the final amount to 1 mL total. The chromatin was then sheared to a median size of ~300 bp by sonication using Covaris S2 (Adaptive Focused Acoustics™ (AFA) technology) at the given parameters: Duty cycle: 20%, Intensity: 5, Cycles/burst: 200, Temp: 7 °C, Time: 6 min. The sonicated lysate was collected, centrifuged at 4000 g for 10 minutes at 4 °C and the supernatant collected was diluted at 1:1 ratio using FA-lysis buffer (50 mM HEPES-KOH pH 7.5, 2 mM EDTA pH 7.5, 150 mM NaCl, 1% TritonX-100, 0.1% Sodium Deoxycholate) to dilute the SDS concentration to 0.05%. The samples were then incubated overnight at 4 °C with 2µg of H3K9me2 antibody (mAbcam 1220). Following antibody binding, the samples were incubated at 4 °C for at least 1 h with ~40 µL of appropriate bead slurry (nProtein A Sepharose, GE Healthcare, 5280-01). Following incubation, the beads were collected by centrifugation at 4 °C at 96 g for 1 min. The beads were then washed twice with washing buffer I (50 mM HEPES-KOH pH 7.5, 1 mM EDTA pH 7.5, 140 mM NaCl, 1% TritonX-100, 0.1% Sodium Deoxycholate) and wash buffer II (50 mM HEPES-KOH pH 7.5, 1 mM EDTA pH 7.5, 0.5 M NaCl, 1% TritonX-100, 0.1% Sodium Deoxycholate). Finally, the beads were washed once again with wash buffer III (10 mM Tris-Cl pH 8, 1 mM EDTA pH 7.5, 250 mM LiCl, 1% NP-40 Igepal, 0.5% Sodium Deoxycholate) and briefly with 1 mL Tris-EDTA pH 8 buffer. The bound protein and chromatin fractions were eluted twice by adding 50 µL of elution buffer (10 mM Tris-HCl pH 8, 10 mM EDTA pH 7.5, 2% SDS) at 65 °C for 10−15 min on a thermomixer. The samples were then de-crosslinked by incubating at 65 °C overnight after addition of 2 µL proteinase K. The DNA was purified using Phenol: Chloroform: Isomyl alcohol method following RNase treatment. The purified DNA pellet was eluted in 15−20 µL of nuclease-free water (Invitrogen, 10977015) or 0.1x TE and was used for library preparation using NEBNext Ultra II DNA library prep kit for Illumina (NEB, E7645) following manufacturer instructions. The final cDNA libraries were amplified using NEBNext Multiplex Oligos for illumina (NEB, E7335) and purified using SPRIselect size selection beads (Beckman Coulter, B23317). Individual library quality was assessed on TapeStation (Agilent 4200) using a DNA D1000 High sensitivity tape. Indexed libraries were pooled and sequenced on NovaSeq sequencer with 150 bp paired-end reads.

## ChIP-seq analysis

Paired-end illumina reads were subject to a thorough Quality Control (QC). Shortly, FastQC 0.11.8[109] as used to generate QC reports. Low complexity regions were filtered out with prinseq-lite.pl from PRINSEQ-lite 0.20.4[110] maximum dust was set to 3) and the remaining sequences were trimmed using Trimmomatic[111] (SLIDINGWINDOW: 4 nt; phred quality cut-off: 15, MINLEN = 36). QC filtered were aligned with BWA MEM −0.7.17[112]. Aligned bam files were sorted and indexed using samtools 1.3.1[104] Potential PCR duplicates were removed with samtools using the rmdup option. Bigwig tracks were generated using bamCoverage[105] to the pombe genome (ASM294v2) with the parameter "−binSize 1". Bigwig tracks were normalised by the sum of the raw signals of chromosome I and II multiplied by 100 million. Potential artifacts were defined as regions consisting of <15 edges and subsequently removed with a custom R script using rtracklayer[113] and GenomicRanges[106] packages. Meta plots were generated using in-house R-based pipeline.

## Generating the Pla1 interaction mutants

The $red1_{W298A/F305A}$ and $red1_{Δ288-345}$ mutants were generated using conventional primer-based cloning. Together with the wild-type Red1, these constructs were cloned into a pFa6a-Hygro plasmid in untagged and 3xFTP tagged versions. The primers used to generate the mutations are listed in Supplementary Table 5. The plasmids generated were confirmed for mutations using PCR and Sanger sequencing at the ACRF Biomolecular Resource Facility at JCSMR, ANU. The desired cassettes were then digested from the positive plasmid using Spe1 restriction enzyme that cuts in the 5′UTR and 3′UTR sequence generating overlapping region for homologous recombination in the Red1 locus when used to transform into a red1Δ strain.

In-order to generate strains retaining the 3′UTR, 300 bp starting from the stop codon of Red1 gene was amplified along with a 500 bp homology region upstream. Similarly, 500 bp region following the 300 bp was amplified and Gibson assembled to PCR amplified product from pFA6a-NatNT2 using appropriate primer pairs. This overrode the Hygromycin resistance marker to a natNT2 resistance along with retaining 300 bp 3′UTR region of Red1. Hence all the untagged mutants used for RNA expression analysis were constructed retaining their 3′UTR region.

## Reporting summary

Further information on research design is available in the Nature Portfolio Reporting Summary linked to this article.

## Data availability

The data that support this study are available from the corresponding authors upon reasonable request. NMR backbone chemical shifts for Red1 have been deposited to the BMRB under the accession code 50680. Coordinates and structure factors for the Pla1-Red1 complex, Pla1$_{FL}$ and Pla1$_{\Delta14}$ have been deposited in the PDB with accession codes 7Q72, 7Q73 and 7Q74, respectively. SAXS data for Pla1-Red1 complex have been deposited to the SASBDB with accession code SASDKE6. Genome-wide datasets are deposited in NCBI GEO under the reference number GEO: GSE206106. Source data are provided with this paper.

## Code availability

Custom R and Bash scripts that were used to analyse the data is available at https://github.com/ahorvath/Soni-et-al.-Nat.-Commun.-2022.

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

## Acknowledgements

We thank Jürgen Kopp and Claudia Siegmann from the BZH/Cluster of Excellence: CellNetworks crystallization platform and acknowledge access to the beamlines P12, P13 and P14 at PETRA III at DESY in Hamburg and the support of the beamline scientists. We thank Per Haberkant and Frank Stein at the Proteomics Core Facility (PCF), EMBL Heidelberg for mass spectrometry analysis. This work was supported by a DAAD fellowship to N.D., the Deutsche Forschungsgemeinschaft (DFG) through the Leibniz program (SI 586/6-1) and TRR 319 (Project-ID 439669440, TP B03) to I.S. and the Australian Research Council's Discovery Projects funding scheme to T.F. (project DP190100423).

## Author contributions

K.S., A.S., T.F. and I.S. designed the study and interpreted the results. K.S. performed NMR, crystallographic structure determination, SAXS, in vitro polyadenylation assay, biophysical characterisation using pulldown assays, ITC, BLI and Y2H. N.D. and L.K. performed Y2H and NMR experiments. A.S. performed TAP purification followed by interaction analysis and mass spectrometry, Illumina sequencing of total, poly(A) + and RIP-ed RNA, Direct RNA sequencing by Nanopore of poly(A) + and RIP-ed RNA, ChIP seq. experiments. R.H. supported Nanopore sequencing experiments. A.H. and T.F. performed the bioinformatics analyses. B.S. recorded NMR experiments. K.W. supported crystallographic analysis. K.S, I.S. and T.F wrote the manuscript. All authors contributed to the final version of the manuscript.

## Competing interests

The authors declare no competing interests.
