## [Peer Review File · Nature Communications]

REVIEWER COMMENTS

Reviewer #1 (Remarks to the Author):

Soni et al characterise the interaction of two *S.pombe* MTREC subunits: the canonical poly(A) polymerase, Pla1, and the zinc finger protein Red1. Using yeast-two hybrid (Y2H) and in vitro pulldown assays they map Red1 and Pla1 binding regions, which are validated by NMR and isothermal titration calorimetry (ITC). Crystal structures of Pla1 in complex with Red1288-345 show that two regions of Red1 contact the Pla1 RRM, and mutational analyses followed by ITC are used to assess each region's importance for binding in vitro. A functional significance of the Red1-Pla1 interaction is addressed by in vitro polyadenylation assays, showing that the presence of Red1288-345 causes a switch from processive to distributive poly(A) addition by Pla1. Poly(A) lengths of Red1-associated CUTs and meiotic mRNAs are measured by direct RNA sequencing to be nearly twice as long as Red1-associated control mRNAs. In the Red1-Pla1 interaction mutants, Red1-interacting CUTs and meiotic mRNA poly(A) lengths are reduced to WT mRNA poly(A) lengths. Curiously, poly(A) length appears to be uncoupled from decay, since these transcripts show either modest (PROMPTs) or no (meiotic RNAs) differences in steady state RNA level when the Red1-Pla1 interaction is disrupted. In light of the overlapping poly(A) polymerase binding interfaces of *S.pombe* Red1 and *S.cerevisiae* Fip1, the authors finally show that Red1 can outcompete the CPF factor Iss1 for binding of Pla1, possibly indicating coordinated action of the two complexes.

Overall, the manuscript is of high quality. It enhances our understanding of both the structural organisation of MTREC, and a potential role of polyadenylation in *S.pombe* nuclear RNA turnover. While the manuscript flows well from figures 1-5, it becomes somewhat disjointed by jumping from polyadenylation to heterochromatic islands and then back to the CPF complex, which could perhaps be remedied by swapping figures 6 and 7. I am in favour of publication in Nature Communications once the following points are addressed.

Major point:

- A concern is that the authors may have missed part of the Red1-Pla1 interaction. From the ITC data, showing the Pla1RRM binding affinities, the interaction is stronger for Red1288-345 as compared to Red1288-322. However, it is not straightforward to understand the actual segment visible in the electron density (please add details what was crystallized and what part of the molecules is present in the model, and also replace 'N' and 'C' in Figure 2B with the actual residue numbers). It seems that there is no model for the part of Red1 following the β -stand (probably stopping at around residue 322). This region includes a negatively charged patch that the authors show by mutagenesis is essential for binding and they speculate how these residues could interact with Pla1. The authors are encouraged to exploit the newest version of AlphaFold for protein complexes to extend their experimentally-derived model of the Pla1- Red1 complex and obtain a more complete model of the interaction.

Other points:

- In lines 30, 78, and 85, the authors refer to the PAXT 'complex', when it has so far been described as a 'connection'
- The introduction switches between organisms throughout, and it would be very hard to follow as a non-expert. One suggestion would be to prefix protein names e.g. spRed1, hsZFC3H1 etc...
- Line 66: 'herculean' is a bit bombastic
- Line 80: is it known that PAXT targets have 'extensive' poly(A) tails?
- Line 93-95: This sentence is not clear and needs re-writing.
- In line 132 and multiple times throughout, the authors refer to PROMPTs, however earlier

in the introduction they imply that PROMPTs are the mammalian term for CUTs

- In figure 1, there is a lack of consistency of style for the domain figures in A and C.
- In figure 1C, it may be informative to indicate the known binding surfaces on Red1 for the other MTREC components and to show a schematic of the whole MTREC complex
- In line 306 and 307: Pla1EEQE mutation led to 3-fold and Pla1EEE mutation led to 11-fold loss in affinity. But the Figure 3E and 3F show affinities of 2.8 and 9.8 μM , which equal 2- and 7-fold, respectively (in Figure 1E the K_D for wild type is 1.4 μM)
- In figure 2B, although a colour scheme is used for the structure, labels for Pla1 and Red1 would make it easier to follow.
- Regarding figure 3 (A-F) it is not clear in the results, figure or legend whether the mutants used are in the context of full length proteins or small peptides. This requires clarification.
- In figure 3G, the authors present data showing that Red1288-345, but not an interaction mutant, can switch the in vitro activity of Pla1 from processive to distributive, however, the implications of this are not discussed, therefore:
 - o Are these data included as further confirmation of the Pla1-Red1 interaction, or do they have relevance to in vivo Pla1 activity? (If the latter is the case, it would be interesting to know whether other Pla1 interactors can also elicit distributive activity....)
 - o It is also not mentioned how many times these experiments were repeated
- The results title relating to figure 4 is rather vague and could be more descriptive
- Very little is said about figure 4B. It may be worth including a comment about the stoichiometry of the components. Is the prominent band Red1?
- In figure 4C, Pab2 and Mmi1 are absent from the heatmap - it is a bit misleading to say that MTREC is otherwise intact without including all the subunits
- It is good to include an IP-western blot to validate these results, however, this would be further appreciated if it were to include a blot for a protein that would not change in the mutant e.g. Mtl1
 - o Strange migration is due to the TEV cleavage step during the tandem purification, but this may not be easy to follow at first glance. It would be helpful to include a schematic of the purification steps above the IP?
 - o The membrane is cut a bit too closely, especially when the $\Delta 288-345$ mutant is right at the bottom of the FLAG blot
- In figure 5 (and in-text) the mixed use of PROMPTs and CUTs is quite confusing to follow, especially as the introduction defines them as mammalian or yeast transcripts respectively. Though this may be in order to distinguish promoter upstream CUTs from 3' intergenic CUTs, the authors could be more clear.
 - 3'IGTs are mentioned but not explained. Are these caused by readthrough, or are they 3' extended products which are usually trimmed?
- In figure 5C, the tracks for sense and antisense are mirrored. As this is not a common format, it makes it hard to follow. It also seems at first glance as though the two panels of C belong to A or B. In addition, it would be useful to highlight the PROMPT/3'IGT of interest as this is not immediately obvious
- Although it is mentioned in the text, it is not clear in the figure or legend that the data in 5 D-E are from RIP samples. Though it is appreciated that the purpose of the RIP was to enrich for RNAs of interest, some quality control to demonstrate comparable IP efficiencies should be available. In addition, does Red1 definitely not bind mRNAs? The authors suggest that they are contaminating transcripts in their RIP experiment, how is this known? Did the authors consider dT purification of nuclear RNA as an alternative/adjunct? Although it misses unadenylated or very short-tailed transcripts, it would allow tail lengths in red1 Δ to be included as a comparison.
- The poly(A) length data in 5 D-E would be better represented as length distributions. Later, in lines 551-552, it is noted that intergenic and long antisense CUTs are harder to define and that the analyses likely include stable ncRNAs which are not hyperadenylated; while separate populations might not necessarily be visible as separate peaks in the tail

length distribution, it will be even more hidden when only comparing summary statistics.

- Lines 383-389 should be moved to the materials and methods section
- Are the genome-wide data mentioned in lines 401-403 from the Red1 IP?
- Lines 413-416 – for meiotic mRNAs at least, the decrease in poly(A) length observed in the mutant strains does not really seem 'less pronounced' than for PROMPTs
- Figure 6A is difficult to read and therefore not that informative. Could it be displayed in an alternative way e.g. heatmaps?
- In figure 6 the authors show that H3K9me2 levels are decreased upon red1 Δ or the Pla1 interacting mutant. The mechanism behind this could be expanded a bit further. Is the recruitment of MTREC to these genomic regions being impaired in the mutants? Can Red1 or Mtl1 be ChIPed in these regions? Also, is the recruitment of Ctr4 reduced in these conditions? Either some ChIPseq or ChIPqPCR would be informative to the model.
- In line 471 , it is written as if Fip1 exists in *S.pombe*. This should be re-phrased.
- A good control in figure 7B would be the Red1 mutant which does not bind Pla1 (as was nicely used in figure 3)
- Line 515 – 'leading to the inefficient degradation of these transcripts' also seems to refer to meiotic mRNAs, but their reduced poly(A) length did not seem to affect their steady state level.
- Lines 539-543 – this sentence is quite long and convoluted and could do with re-phrasing and avoiding the use of 'don't'
- Line 544 – it is written that the authors found the median poly(A) tail length of mRNAs in WT *S.pombe* to be 32 nts, but doesn't this measurement only include the mRNAs which were pulled down in the Red1 RIP? Is a median length of 32 nt expected given previous poly(A) length determinations of ~ 50-80 adenosines for fission yeast mRNAs?
- Lines 578-581 – The mention of NEXT subunits co-purifying with 3' processing machinery is a bit strange, especially when PAXT is not brought up. The referenced study used reporter RNAs based on viral poly(A) signals to pull down associated proteins in HeLa nuclear extract in the presence of ATP, and recovered a large number of nuclear proteins. In the supplementary table S2 of that paper, IPs using the 3' processing factors CPSF73 and CstF77 did not co-purify NEXT or exosome subunits.

Reviewer #2 (Remarks to the Author):

This is an exceptionally complete, multidisciplinary study of the interaction of *S. pombe* Pla1 (a poly(A) polymerase) with Red1, a core component of the MTREC complex. In addition to using a variety of structural and biophysical approaches (X-ray crystallography, structural NMR, SAXS, ITC, and more) the authors employ yeast genetics, gel-based polyadenylation assays and whole-genome RNA analysis to round out their story of how polyadenylation contributes to RNA surveillance. The manuscript is clear and well-written, and the sequence in which the various experiments are presented is exemplary. This work represents a significant step forward in our understanding of this complex process as well as of some of the most important molecular details. I believe this manuscript can be accepted as is. I congratulate the authors on the quality of their work.

Reviewer #3 (Remarks to the Author):

The present study by Soni et al. reports a structure-function study of the Pla1-Red1 complex in the context of RNA surveillance. Pla1 is a poly(A) polymerase usually part of the highly conserved cleavage and polyadenylation factor (CPF), whereas Red1 is the central scaffolding protein of the MTREC complex responsible for degradation of cryptic unstable transcripts (CUTs). Interaction between Pla1 and Red1 was previously proposed based on

co-localisation and co-immunoprecipitation. Here the authors, using a combination of NMR, crystallography, ITC, SAXS and biochemical assays in combination with RNA sequencing experiments analysis, report on the structure of the Pla1-Red1 complex (an unstructured segment of Red1 interacts with the RRM domain of Pla1) and provide a detailed analysis of the functional implications of the solved structure in the context of RNA degradation.

The manuscript is well-written and the data support the conclusions and claims. Thus, the work is certainly suitable for publication in Nature Communications since the results give a deeper understanding of MTREC complex functioning.

I have only one major point and few minor points that should be addressed before publishing.

Major points:

1) Table 1: For the Pla1-Red1 structure, I do not understand why if 1049 unique reflections were present in the data set for the highest resolution shell, only 654 were used in the refinement procedure. From the 20986-20527 = 459 reflections not used in refinement, the vast majority (ie 1049-654 = 395) are in the highest resolution shell. Is there a problem here on how the anisotropically scaled data from STARANISO were used? Does this structure need to be adjusted/re-refined against the correct dataset?

Minor points:

2) Figure 1A: please draw the different domains at scale. In particular, the unstructured C-terminus appears quite too long.

3) Figure 1C and text l. 161-164: It is not clear how the boundaries of the different fragments were chosen. Please explain any rationale behind these choices.

4) Figure 1, 3, 7 and Figure S6: For the ITC data, and for ease of comparison, please plot all the ITC experimental raw data (DP vs. time) with the same scale on the ucal/s axis. Please also provide the N values (number of sites). Explain whether N was fixed or adjusted during the fitting procedures?

5) Figure 1, 3, 7 and Figure S6: For the ITC data, instead of reporting only the average values of the two independent experiments, and to comply with the journal requirements ("individual data points are shown when possible, and always for $n < 10$ "), please report in a Supplementary Table the K_d and N values of each individual ITC experiment.

6) l. 167-169: "A two dimensional 1H , ^{15}N -HSQC spectrum of Red1288-322 shows that this region of Red1 is largely unstructured, as inferred from the low chemical shift dispersion in the 1H -dimension (Figure 1D)." Is this fragment also unstructured in a broader context? Is it clear that additional elements are not missing for the fragment to fold? If we take this reasoning to the extreme, is this fragment really unstructured in the full length Red1?

7) l.183-184: "We found that while Red1288-322 binds to Pla1 with a $K_D = 7.9 \mu M$ (Figure 1E), the addition of residues 323-345 in Red1288-345 leads to a ~ 5.6 -fold increase in binding affinity ($K_D = 1.4 \mu M$, Figure 1F). We therefore refer to the Red1 fragment comprising residues 288-345 as the complete Pla1 interaction site." It does not mean that the 288-345 fragment is the complete Pla1 interaction site. It is the best among the few that were tested, but maybe larger fragments bind even better. Please revise the statement or explain further.

8) I. 228-229: "Co-crystals of the complex appeared within three weeks and diffracted to 2.81 Å resolution." In the material and method section as well as in table 1, it is mentioned that the co-crystal showed anisotropic diffraction. The resolution along the best diffracting axis is given, but it would be also important for the reader to get an idea about the quality of the diffraction to provide as well the approximate resolution along the two other axes.

9) Figure 2B,C and text I. 249-252: "The first half comprises a short α -helix at the N-terminus (a0) that helps orient Red1 onto Pla1 (Figure 2C). A network of aromatic stacking interactions occurs between Pla1 RRM Phe468 and Red1 Trp298 and Phe305 (Figure 2C, Supplementary Figure S5C)." Is there a tendency for this region of Red1 to adopt transient α -helical structures in its free form? Please report on the degree of structuration of the Red1 fragment 288-322, estimated from NMR chemical shifts and their derived secondary structure propensity.

10) Figure S5H and I. 274-276: "Subsequent fitting of the experimental SAXS scattering profile of the complex with the crystal structure show that the data are in good agreement with a χ^2 fitting-value of 1.23 (Supplementary Figure S5H)." To better appreciate the quality of the agreement between the scattering data and the fitting curve, please provide the fitting residuals.

11) I. 307-309: "indicating that these charged patches of Pla1 and Red1 C-termini are in close proximity in solution and are involved in electrostatic interactions." Plausible explanation indeed, but please be more cautious with the interpretation here. Mutations could also affect the overall structure and re-arrangements can impact the affinity even though these regions are not directly involved.

12) I. 340: "mass spectrometry" not "mass spectroscopy".

13) Figure 7 and text I. 484-488: Are the KDs derived from the the k_{on} and k_{off} values measured with BLI compatible/coherent with the one measured with ITC? Please make the calculation and provide comments on the comparison.

14) Figure 8: Please explain the meaning of the the "?" and "??" signs on the model. It is not clear to what sentence/idea they are exactly referring to in the figure legend.

15) I. 685-687: For SAXS data, it is important for the sample to be homogeneous. How did the authors make sure that the formed complex is really a 1:1 complex and that none of the partners are in slight excess, to ensure a flawless SAXS data acquisition?

REVIEWER COMMENTS

We thank all of the reviewers for their positive feedback and insightful comments. Below we have addressed each of the reviewer's comments. The original reviewer comments are included in blue text, while our responses are in black.

Reviewer #1 (Remarks to the Author):

Soni et al characterise the interaction of two *S.pombe* MTREC subunits: the canonical poly(A) polymerase, Pla1, and the zinc finger protein Red1. Using yeast-two hybrid (Y2H) and in vitro pulldown assays they map Red1 and Pla1 binding regions, which are validated by NMR and isothermal titration calorimetry (ITC). Crystal structures of Pla1 in complex with Red1288-345 show that two regions of Red1 contact the Pla1 RRM, and mutational analyses followed by ITC are used to assess each region's importance for binding in vitro. A functional significance of the Red1-Pla1 interaction is addressed by in vitro polyadenylation assays, showing that the presence of Red1288-345 causes a switch from processive to distributive poly(A) addition by Pla1. Poly(A) lengths of Red1-associated CUTs and meiotic mRNAs are measured by direct RNA sequencing to be nearly twice as long as Red1-associated control mRNAs. In the Red1-Pla1 interaction mutants, Red1-interacting CUTs and meiotic mRNA poly(A)

lengths are reduced to WT mRNA poly(A) lengths. Curiously, poly(A) length appears to be uncoupled from decay, since these transcripts show either modest (PROMPTs) or no (meiotic RNAs) differences in steady state RNA level when the Red1-Pla1 interaction is disrupted. In light of the overlapping poly(A) polymerase binding interfaces of *S.pombe* Red1 and *S.cerevisiae* Fip1, the authors finally show that Red1 can outcompete the CPF factor Iss1 for binding of Pla1, possibly indicating coordinated action of the two complexes.

Overall, the manuscript is of high quality. It enhances our understanding of both the structural organisation of MTREC, and a potential role of polyadenylation in *S.pombe* nuclear RNA turnover. While the manuscript flows well from figures 1-5, it becomes somewhat disjointed by jumping from polyadenylation to heterochromatic islands and then back to the CPF complex, which could perhaps be remedied by swapping figures 6 and 7. I am in favour of publication in Nature Communications once the following points are addressed.

We thank the reviewer for the positive comments and also for the specific points raised. We have implemented most of the suggestions (see below in the text) and by doing so, we believe that our manuscript has improved. As suggested, we swapped the order of figures 6 and 7.

Major point:

- A concern is that the authors may have missed part of the Red1-Pla1 interaction. From the ITC data, showing the Pla1RRM binding affinities, the interaction is stronger for Red1288-345 as compared to Red1288-322. However, it is not straightforward to understand the actual segment visible in the electron density (please add details what was crystallized and what part of the molecules is present in the model, and also replace 'N' and 'C' in Figure 2B with the actual residue numbers). It seems that there is no model for the part of Red1 following the β -stand (probably stopping at around residue 322). This region includes a negatively charged patch that the authors show by mutagenesis is essential for binding and they speculate how these residues could interact with Pla1. The authors are encouraged to exploit the newest version of AlphaFold for protein complexes to extend their experimentally-derived model of the Pla1- Red1 complex and obtain a more complete model of the interaction.

We have included a new panel in Supplementary Figure S6 (S6D), which shows the AlphaFold2 based model of the Pla1-Red1 complex. We have also introduced a corresponding statement in the main text: "In accordance with these data, structure prediction using ColabFold also shows that Pla1 residues K544/K545/R546 are spatially close to Red1 residues D317/S318/D319/D320 (**Supplementary Figure S6D**)". However, we would not go so far as to extend the experimental model using the predictions since the per-residue confidence score in the C-terminal tails of Pla1 and Red1 is quite low and the 5 models that were generated do not really converge at the C-terminus of Red1 (see figure below).

Figure for Reviewer 1: Structure alignment of the five Pla1-Red1 models generated using ColabFold. For ease of comparison, only Pla1 from the top model is shown (cyan). Red1 from the different models is coloured in wheat (top model), orange, pink, blue and green; the N- and C-termini are marked.

Other points:

- In lines 30, 78, and 85, the authors refer to the PAXT 'complex', when it has so far been described as a 'connection'

Changed to PAXT connection.

- The introduction switches between organisms throughout, and it would be very hard to follow as a non-expert. One suggestion would be to prefix protein names e.g. spRed1, hsZFC3H1 etc...

We prefixed all protein/gene names that are not *S. pombe* proteins/genes.

- Line 66: 'herculean' is a bit bombastic

Agreed, deleted.

- Line 80: is it known that PAXT targets have 'extensive' poly(A) tails?

"Extensive" deleted.

- Line 93-95: This sentence is not clear and needs re-writing.

Sentence changed.

- In line 132 and multiple times throughout, the authors refer to PROMPTs, however earlier in the introduction they imply that PROMPTs are the mammalian term for CUTs

In yeast, CUTs are defined as transcripts that are stabilized in nuclear exosome mutant *rrp6Δ*. Promoter associated antisense transcripts (PROMPTs) are a distinct subtype of CUTs, among other CUT subtypes, such as antisense transcripts (within ORFs) and intergenic transcripts that are not associated with promoter regions (3'IGTs). Nuclear exosome in mammalian cells is also responsible for the degradation of various nuclear RNAs, including mammalian PROMPTs. In our opinion, it would be useful to have a unifying nomenclature for all these short half-life-time nuclear transcripts and call these all together CUTs – similar to yeast, but the current nomenclature in mammalian literature doesn't use the term "CUTs".

- In figure 1, there is a lack of consistency of style for the domain figures in A and C.

We changed the figure as suggested.

- In figure 1C, it may be informative to indicate the known binding surfaces on Red1 for the other MTREC components and to show a schematic of the whole MTREC complex
We tried various versions for this figure but adding this information to Fig 1C made the panel very crowded and drew attention away from the main message. We have therefore included a new supplemental panel (Supplementary Figure S1A) to indicate the known binding surfaces, as suggested.

- In line 306 and 307: Pla1EEQE mutation led to 3-fold and Pla1EEE mutation led to 11-fold loss in affinity. But the Figure 3E and 3F show affinities of 2.8 and 9.8 μ M, which equal 2- and 7-fold, respectively (in Figure 1E the KD for wild type is 1.4 μ M)
We apologize for this error. It has been corrected in the text.

- In figure 2B, although a colour scheme is used for the structure, labels for Pla1 and Red1 would make it easier to follow.
Done.

- Regarding figure 3 (A-F) it is not clear in the results, figure or legend whether the mutants used are in the context of full length proteins or small peptides. This requires clarification.
We used Pla1 RRM domain and Red1₂₈₈₋₃₄₅ for these experiments. This is now clarified in the text.

- In figure 3G, the authors present data showing that Red1₂₈₈₋₃₄₅, but not an interaction mutant, can switch the *in vitro* activity of Pla1 from processive to distributive, however, the implications of this are not discussed, therefore:

o Are these data included as further confirmation of the Pla1-Red1 interaction, or do they have relevance to *in vivo* Pla1 activity? (If the latter is the case, it would be interesting to know whether other Pla1 interactors can also elicit distributive activity....)

We include these data to show that *in vitro* purified Pla1 is enzymatically active and also serves as a further confirmation of Pla1-Red1 interaction *in vitro*. Interestingly, scPap1 becomes more distributive in *in vitro* polyadenylation assays upon scFip1 binding (Helmling et al., 2001; Zhelkovsky et al., 1998), similar to our results with Red1-bound Pla1, although this seems to be contra-intuitive in *in vivo* settings. The processivity of Pla1 and the effect of Red1 binding is probably different *in vivo*, when the RNA is tethered to the MTREC complex not only by Pla1 but also by other components of MTREC (including the helicase Mtl1) that also have RNA binding capacity.

o It is also not mentioned how many times these experiments were repeated Figure legend
Representative gels from two independent experiments are shown. A statement has been added to the Figure legend 3G.

- The results title relating to figure 4 is rather vague and could be more descriptive
We have changed the title of Figure 4.

- Very little is said about figure 4B. It may be worth including a comment about the stoichiometry of the components. Is the prominent band Red1?

We did not indicate the protein names next to the gel because we did not determine the individual bands from these particular purifications, however, based on our previous purifications where individual bands were identified, the prominent band at the top is Mtl1, 2nd band from top is contamination, the 3rd band from the top is Red1 (or Red1 mutants) which is the bait in these purifications.

We previously showed (Zhou et al. 2015) that various components/submodules of the complex have different stoichiometry, suggesting that MTREC complex might exist in various forms or multiple development stages. Our split-tag purifications (and other data) confirmed that the "super-complex" with all 11 subunits is present in the cells (Zhou et al. 2015), justifying the use of the term "complex" in the context of MTREC complex.

We would like to avoid speculating about stoichiometry based on these gels and mass spectrometry results because they are not reliable measures to conclude stoichiometry within the complex (or complex mixture). Further biochemical and structural work will be necessary to draw solid conclusions about the overall architecture of this large and dynamic complex.

- In figure 4C, Pab2 and Mmi1 are absent from the heatmap - it is a bit misleading to say that MTREC is otherwise intact without including all the subunits

In the heat map in Figure 4C we originally included only the proteins that directly interact with Red1. We have modified Fig 4C and extended the heatmap with all MTREC complex subunits that were reliably quantified by our TMT 10plex mass spectrometry analysis, including Pab2 and Mmi1.

- It is good to include an IP-western blot to validate these results, however, this would be further appreciated if it were to include a blot for a protein that would not change in the mutant e.g. Mtl1

Both Fig 4B and 4C show that the amount of co-purifying Mtl1 (in relation to the amount of purified Red1 bait) is unaffected by the Red1 mutations (Red1_{W298A/F305A} and Red1 _{Δ 288-345}).

o Strange migration is due to the TEV cleavage step during the tandem purification, but this may not be easy to follow at first glance. It would be helpful to include a schematic of the purification steps above the IP?

We included the following sentence in the figure legend to explain the migration pattern of the Red1 bait protein: "Note that the Red1 bait protein in the Flag eluate runs ~20kDa lower than in the input, due to the cleavage of the C-terminal Protein-A tag by the TEV enzyme during the elution step from the IgG column (see details in Methods section)."

o The membrane is cut a bit too closely, especially when the $\Delta 288-345$ mutant is right at the bottom of the FLAG blot

Since we use the same gel/membrane to probe for the bait protein and the co-purifying Pla1 protein (to stick to the highest standard in these experiments) we had to cut the membrane at ~100kDa to avoid getting too close to the Pla1-HA, which runs at ~90kDa. We included the following sentence in the figure legend to explain why the membrane was cut relatively close to the Red1 $\Delta 288-345$ mutant: "Inputs and final eluates of the tandem affinity purifications of the indicated strains were blotted to nitrocellulose membrane and cut around the 100kDa marker. The bottom part of the membrane was probed with α -HA to detect co-purifying Pla1-HA (top panel) and the top part was probed with α -Flag to detect the bait protein Red1-FTP (bottom panel)."

- In figure 5 (and in-text) the mixed use of PROMPTs and CUTs is quite confusing to follow, especially as the introduction defines them as mammalian or yeast transcripts respectively. Though this may be in order to distinguish promoter upstream CUTs from 3' intergenic CUTs, the authors could be more clear.

We are using these terms consequently, as introduced in our introduction, and also accepted in yeast literature: CUTs include all transcripts that are stabilized in nuclear exosome mutants (and also in *red1* Δ), the 3 main categories of these are: 1) PROMPTs; 2) antisense transcripts (AS); 3) 3' intergenic transcripts (3'IGTs). In *S. pombe*, a large portion of meiotic mRNAs are also stabilized in nuclear exosome mutants (and in *red1* Δ), but the literature doesn't generally refer to them as CUTs. The discrepancy between the mammalian nomenclature and the yeast nomenclature is somewhat disturbing, but mammalian nomenclature does not have an equivalent name for intergenic and antisense CUTs. It is unclear for us if these CUT subclasses are not present in mammalian nuclear exosome mutants or if they are not yet characterized.

- 3'IGTs are mentioned but not explained. Are these caused by readthrough, or are they 3' extended products which are usually trimmed?

Both 3' intergenic transcripts (3'IGTs) and antisense transcripts might be the products of read-through transcription, but this has not been shown and there are also other possibilities to explain how these transcripts might originate. The fact is that these transcripts are strongly stabilized (or upregulated) in the *rrp6* Δ strain (this is the reason they are defined as CUTs), but also in MTREC mutants, such as in *red1* Δ or *pab2* Δ strains. We are not aware if it was clearly shown if these transcripts are stabilized transcripts or if they are the product of a termination defect in these mutants. These questions are interesting and important but go beyond the scope of our manuscript.

- In figure 5C, the tracks for sense and antisense are mirrored. As this is not a common format, it makes it hard to follow. It also seems at first glance as though the two panels of C belong to A or B. In addition, it would be useful to highlight the PROMPT/3'IGT of interest as this is not immediately obvious

We have re-shuffled the panels as suggested and highlighted a representative PROMPT and AS transcript. The order of the tracks (mirrored around the reference genome or not mirrored) is a question of preference and we prefer to keep it in the current format.

- Although it is mentioned in the text, it is not clear in the figure or legend that the data in 5 D-E are from RIP samples. Though it is appreciated that the purpose of the RIP was to enrich for RNAs of interest, some quality control to demonstrate comparable IP efficiencies should be available. In addition, does Red1 definitely not bind mRNAs? The authors suggest that they are contaminating transcripts in their RIP experiment, how is this known? Did the authors consider dT purification of nuclear RNA as an alternative/adjunct? Although it misses unadenylated or very short-tailed transcripts, it would allow tail lengths in *red1* Δ to be included as a comparison.

- We have modified the figure legend for Figure 5D,E, clearly stating that these plots represent MTREC associated RNA transcripts.

- In general, the majority of the IP-ed RNA or DNA in RIP or ChIP experiments are background contaminants and represent the total transcriptome/genome. Specific signal is enriched over this background signal and often negligible in amount compared to the background RNA/DNA amount. It is extremely difficult to exclude that a portion of these mRNAs is specifically binding to the MTREC complex, but mRNAs in general are not enriched in Red1 RIP experiments (except from meiotic mRNAs), they are fully spliced and processed and very likely represent the general background of the RIP technique. Since we cannot formally exclude that part of the IP-ed mRNAs represent specific binding, we included the word "likely" in the description of these mRNAs ("likely representing contaminating RNA transcripts in these RIP experiments"). However, CUTs are enriched in the Red1 IP fraction (both in WT and in our mutants, see figure below). We previously published in Zhou et al. 2015, that RIP experiments using MTREC subunits as bait, enrich CUTs and meiotic mRNAs over mRNA transcripts. However, from the particular RIP experiments presented in Fig. 5 D,E we confirmed CUT enrichment by sequencing small aliquots of IP-ed RNA and total RNA (we used Illumina RNA-seq libraries to test this). Figures

below show IP-ed RNA (grey, yellow and light-blue lines) and total RNA (black, orange and dark-blue lines) geometric average transcript levels in sense direction (left panel) or antisense direction (right panel) over all transcription units. Data was normalized to show equivalent sense median mRNA levels between strains in IP-ed and in total RNA. The figure shows that IP-ed RNA samples are enriched in intergenic and antisense transcripts both in WT and mutant strains, indicating that IP efficiencies are comparable.

Whether all mRNAs in these RIP experiments are background binding or some of them are specific (in addition to specifically bound meiotic mRNAs), the length of their poly(A) tail does not change in the Pla1 truncated MTREC mutants, while poly(A)-tail length of IP-ed CUTs are different between WT and mutant strains, which is the main message of Figure 5D,E.

We have not tried to establish nuclear isolation from *S. pombe* cells - it is a complicated procedure in fission yeast since it has to go through cell wall digestion (protoplast preparation) before the nuclei can be isolated. This is a lengthy procedure and a very strong stress-signal for the usually elongated cells that all become rounded during protoplast preparation, and it is not known how this affects the transcriptome, especially transcripts that can be stabilized by stress, such as meiotic mRNAs and other CUTs.

- The poly(A) length data in 5 D-E would be better represented as length distributions. Later, in lines 551-552, it is noted that intergenic and long antisense CUTs are harder to define and that the analyses likely include stable ncRNAs which are not hyperadenylated; while separate populations might not necessarily be visible as separate peaks in the tail length distribution, it will be even more hidden when only comparing summary statistics.

As suggested by the reviewer, we included distribution plots (violin plots) in the Supplementary Figure S8 as 4 new panels (Panel C,D,G,H). Indeed, these distribution plots confirm the reviewer suggestion that we likely have separate populations of AS RNAs and also PROMPTS. It is unclear if these separate populations represent misannotated PROMPTS and AS RNAs or differently processed transcripts. Unfortunately, due to the very dense yeast genome, our self-made CUT "annotations" have high uncertainty and very likely contain many stable transcripts that overlap with the genomic locations of CUTs.

- Lines 383-389 should be moved to the materials and methods section

Since this relatively new method is not yet often used, we believe that keeping these few lines in the main text will help the reader to understand the basic concept of how this method estimates poly(A)-tail length and help the better understanding of our results.

- Are the genome-wide data mentioned in lines 401-403 from the Red1 IP?

Yes, these lines refer to the poly(A) length data in Figure 5D, we now refer to the figure in this sentence.

- Lines 413-416 – for meiotic mRNAs at least, the decrease in poly(A) length observed in the mutant strains does not really seem 'less pronounced' than for PROMPTS

"less pronounced" has been deleted.

- Figure 6A is difficult to read and therefore not that informative. Could it be displayed in an alternative way e.g. heatmaps?

Thanks for the suggestion, we replaced these genome-wide views with heat-maps (Figure 7A).

- In figure 6 the authors show that H3K9me2 levels are decreased upon *red1*Δ or the Pla1 interacting mutant. The

mechanism behind this could be expanded a bit further. Is the recruitment of MTREC to these genomic regions being impaired in the mutants? Can Red1 or Mtl1 be ChIPed in these regions? Also, is the recruitment of Clr4 reduced in these conditions? Either some ChIPseq or ChIPqPCR would be informative to the model.

Previous publications (e.g. Vo et al., 2019 and Zofall et al., 2012) showed that MTREC complex is recruited to these meiotic genes through the Mmi1 subunit, that binds DSR sequence motifs on the nascent mRNAs that are transcribed from these genomic loci. We know that the recruitment of the Pla1 truncated MTREC complex to these meiotic mRNAs is not affected because the degradation of these mRNAs is not affected. Clr4 is the only H3K9 methyl transferase in *S. pombe* and as such, it must be recruited to these facultative heterochromatic islands (also, deletion of Clr4 completely eliminates these H3K9me2 islands), but the level of Clr4 is extremely low and/or transient and/or happens only in a small subpopulation of the cells and CHIP results can't show conclusively Clr4 enrichment at these loci. However, it is unclear how Clr4 is recruited to these sites and why only to these sites (and not to other MTREC targets sites) and how the CPF complex and Pla1 is involved in this process. It would be fascinating to understand more about the mechanistic details of this exciting interaction between RNA processing/surveillance and the epigenetic machinery but it is beyond the scope of our current manuscript.

- In line 471, it is written as if Fip1 exists in *S.pombe*. This should be re-phrased.

This sentence has been changed to: "Since the binding interfaces on the polymerase for scFip1 and Red1 are overlapping, we were wondering if Red1 and Iss1, the fission yeast ortholog of scFip1, might compete for binding to Pla1."

- A good control in figure 7B would be the Red1 mutant which does not bind Pla1 (as was nicely used in figure 3)

We thank the reviewer for pointing this out. We have included the control experiment (Supplementary Figure S9C). While Iss1 cannot outcompete wild-type Red1 from the Pla1-Red1 complex, it can bind to Pla1 when the Red1^{W298A/F305A} double mutant is used for complex formation albeit with a slightly weaker affinity ($K_D=4.7 \mu\text{M}$ compared to $2.5 \mu\text{M}$, Supplementary Figure S9C and Figure 7A).

- Line 515 – 'leading to the inefficient degradation of these transcripts' also seems to refer to meiotic mRNAs, but their reduced poly(A) length did not seem to affect their steady state level.

This sentence has been changed to: "... but interestingly, this only leads to the inefficient degradation of PROMPTS."

- Lines 539-543 – this sentence is quite long and convoluted and could do with re-phrasing and avoiding the use of 'don't'

This sentence has been changed.

- Line 544 – it is written that the authors found the median poly(A) tail length of mRNAs in WT *S.pombe* to be 32 nts, but doesn't this measurement only include the mRNAs which were pulled down in the Red1 RIP? Is a median length of 32 nt expected given previous poly(A) length determinations of ~ 50-80 adenosines for fission yeast mRNAs?

A recent publication (Montañés et al., 2022) that also used direct RNA sequencing with nanopore technology to determine poly(A)-tail length of poly(A)+ RNAs from *S. pombe* cells, concluded that median poly(A) tail length of mRNAs is 48.9 nucleotides. Highly expressed transcripts have less than 40nt of median poly(A) tail lengths while lowly expressed transcripts have longer poly(A) tails. The discrepancy between these numbers (~31 versus ~49nts) might arise from the combination of the following factors:

- The length of the poly(A)-tail is an estimate in all these techniques, using the lengths of the low-variance signal at the beginning of each read. Using different bioinformatic tools (such as in the mentioned study versus our study) might result in differences in the absolute numbers, however, the relative differences within a dataset, using the same method with biological replicate datasets, is highly reproducible.
- Montañés et al. used poly(A)+ RNA isolated directly from *S. pombe* cultures, while we used RNA isolated from RNA IPs, which is a long (~24h) procedure, mostly at 4C but without denaturing conditions. It is possible and likely that residual RNase activity shortens poly(A) tail length in our experimental conditions. However, the difference between CUTs and mRNAs and the lack of this difference between WT and Red1 mutant cells can not be explained by differential non-specific RNA degradation during the IP process.
- The sequencing depth in our experiments is much lower than in Montañés et al. which might skew our dataset towards higher expressed mRNAs that have shorter poly(A)-tails.

We included a short paragraph to our discussion to reference Montañés et al. and to discuss potential differences in the experimental conditions and bioinformatic methods.

- Lines 578-581 – The mention of NEXT subunits co-purifying with 3' processing machinery is a bit strange, especially when PAXT is not brought up. The referenced study used reporter RNAs based on viral poly(A) signals to pull down associated proteins in HeLa nuclear extract in the presence of ATP, and recovered a large number of nuclear proteins. In the supplementary table S2 of that paper, IPs using the 3' processing factors CPSF73 and CstF77 did not co-purify NEXT or exosome subunits.

We modified this sentence and took out the reference for NEXT subunits co-purifying with 3' processing factors in mammalian cells.

Reviewer #2 (Remarks to the Author):

This is an exceptionally complete, multidisciplinary study of the interaction of *S. pombe* Pla1 (a poly(A) polymerase) with Red1, a core component of the MTREC complex. In addition to using a variety of structural and biophysical approaches (X-ray crystallography, structural NMR, SAXS, ITC, and more) the authors employ yeast genetics, gel-based polyadenylation assays and whole-genome RNA analysis to round out their story of how polyadenylation contributes to RNA surveillance. The manuscript is clear and well-written, and the sequence in which the various experiments are presented is exemplary. This work represents a significant step forward in our understanding of this complex process as well as of some of the most important molecular details. I believe this manuscript can be accepted as is. I congratulate the authors on the quality of their work.

We thank the reviewer for appreciating our work and for acknowledging our efforts. We are considering framing this remark and hanging it at a prominent place in our office.

Reviewer #3 (Remarks to the Author):

The present study by Soni et al. reports a structure-function study of the Pla1-Red1 complex in the context of RNA surveillance. Pla1 is a poly(A) polymerase usually part of the highly conserved cleavage and polyadenylation factor (CPF), whereas Red1 is the central scaffolding protein of the MTREC complex responsible for degradation of cryptic unstable transcripts (CUTs). Interaction between Pla1 and Red1 was previously proposed based on co-localisation and co-immunoprecipitation. Here the authors, using a combination of NMR, crystallography, ITC, SAXS and biochemical assays in combination with RNA sequencing experiments analysis, report on the structure of the Pla1-Red1 complex (an unstructured segment of Red1 interacts with the RRM domain of Pla1) and provide a detailed analysis of the functional implications of the solved structure in the context of RNA degradation.

The manuscript is well-written and the data support the conclusions and claims. Thus, the work is certainly suitable for publication in Nature Communications since the results give a deeper understanding of MTREC complex functioning.

We thank the reviewer for the positive response and for the constructive comments.

I have only one major point and few minor points that should be addressed before publishing.

Major points:

1) Table 1: For the Pla1-Red1 structure, I do not understand why if 1049 unique reflections were present in the data set for the highest resolution shell, only 654 were used in the refinement procedure. From the 20986-20527 = 459 reflections not used in refinement, the vast majority (ie 1049-654 = 395) are in the highest resolution shell. Is there a problem here on how the anisotropically scaled data from STARANISO were used? Does this structure need to be adjusted/re-refined against the correct dataset?

We thank the referee for spotting this. The reason for this discrepancy (in the number of reflections used in the refinement in the highest resolution shell) is the difference in the definition of the resolution shells between the two programs STARANISO and PHENIX. While STARANISO defines the highest resolution shell ranging from 2.955-2.805 Å, the one from PHENIX is defined as 2.905-2.805 Å. All the data with asterisks * are derived from STARANISO and without * from PHENIX. We have now added the resolution range reported from STARANISO to the Table 1.

Minor points:

2) Figure 1A: please draw the different domains at scale. In particular, the unstructured C-terminus appears quite too long.

We implemented the suggested changes.

3) Figure 1C and text I. 161-164: It is not clear how the boundaries of the different fragments were chosen. Please explain any rationale behind these choices.

Since the Red1 fragment comprising residues 240-345 is rather unstructured, we wanted to test several shorter overlapping fragments. Accordingly, the text has been modified and a new figure panel has been added (Supplementary Fig 1C)- 'Red1 is predicted as rather unstructured in the originally identified Pla1 interacting fragment comprising residues 240-345 (Red1₂₄₀₋₃₄₅) (**Supplementary Figure S1C**). Therefore, we used partially overlapping, short fragments of Red1 comprising residues 259-288 (Red1₂₅₉₋₂₈₈), 288-345 (Red1₂₈₈₋₃₄₅), and 288-322 (Red1₂₈₈₋₃₂₂) in addition to Red1₂₄₀₋₃₄₅.'

4) Figure 1, 3, 7 and Figure S6: For the ITC data, and for ease of comparison, please plot all the ITC

experimental raw data (DP vs. time) with the same scale on the ucal/s axe. Please also provide the N values (number of sites). Explain whether N was fixed or adjusted during the fitting procedures?

The heat generated from the injection of Red1 mutants (or Iss1) into Pla1^{RRM} or Pla1 mutants with wild-type Red1 is quite different between the different samples. We therefore think it is not helping the reader in understanding the data to plot them all on the same scale with DP vs time, especially as the range is from 0.9 ucal/s (Fig1F) to 0.1 ucal/s (Fig3B) for the different samples.

However, we thank the reviewers for the comment on the N-values. We have now included a Supplementary table S1 showing the N values and dissociation constants from individual measurements.

5) Figure 1, 3, 7 and Figure S6: For the ITC data, instead of reporting only the average values of the two independent experiments, and to comply with the journal requirements ("individual data points are shown when possible, and always for $n < 10$ "), please report in a Supplementary Table the Kd and N values of each individual ITC experiments.

Please see above.

6) l. 167-169: "A two dimensional ¹H, ¹⁵N-HSQC spectrum of Red1²⁸⁸⁻³²² shows that this region of Red1 is largely unstructured, as inferred from the low chemical shift dispersion in the ¹H-dimension (Figure 1D)." Is this fragment also unstructured in a broader context? Is it clear that additional elements are not missing for the fragment to fold? If we take this reasoning to the extreme, is this fragments really unstructured in the full length Red1?

Red1 is a quite large (79.5 kDa) and rather unstructured protein, so we have not looked into the structure of the Red1 fragment 288-322 in the context of the full-length protein experimentally. However, now exploiting the availability of AlphaFold based near accurate structure prediction, we can say that Red1 is unstructured in the region of 288-322 also in the context of the full-length protein. Here we provide for the referee the structure prediction, where the Red1 residues 288-345 are coloured in salmon.

7) I.183-184: "We found that while Red1288-322 binds to Pla1 with a $K_D = 7.9 \mu\text{M}$ (Figure 1E), the addition of residues 323-345 in Red1288-345 leads to a ~5.6-fold increase in binding affinity ($K_D = 1.4 \mu\text{M}$, Figure 1F). We therefore refer to the Red1 fragment comprising residues 288-345 as the complete Pla1 interaction site." It does not mean that the 288-345 fragment is the complete Pla1 interaction site. It is the best among the few that were tested, but maybe larger fragments bind even better. Please revise the statement or explain further.

We have edited the statement to 'Therefore, from the fragments tested in this study, the fragment comprising residues 288-345 binds best to Pla1, and is referred to as the Pla1 interaction region.'

8) I. 228-229: "Co-crystals of the complex appeared within three weeks and diffracted to 2.81 Å resolution." In the material and method section as well as in table 1, it is mentioned that the co-crystal showed anisotropic diffraction. The resolution along the best diffracting axis is given, but it would be also important for the reader to get an idea about the quality of the diffraction to provide as well the approximate resolution along the two other axes.

We have included a statement in the Methods section under X-ray crystallography. The diffraction limits for the other axes are 3.2 Å and 3.8 Å.

9) Figure 2B,C and text I. 249-252: "The first half comprises a short α -helix at the N-terminus (α_0) that helps orient Red1 onto Pla1 (Figure 2C). A network of aromatic stacking interactions occurs between Pla1 RRM Phe468 and Red1 Trp298 and Phe305 (Figure 2C, Supplementary Figure S5C)." Is there a tendency for this region of Red1 to adopt transient α -helical structures in its free form? Please report on the degree of structuration of the Red1 fragment 288-322, estimated from NMR chemical shifts and their derived secondary structure propensity.

We included Fig S2 panels A/B showing NMR derived secondary chemical shifts and the secondary structure propensity using TALOS+.

10) Figure S5H and I. 274-276: "Subsequent fitting of the experimental SAXS scattering profile of the complex with the crystal structure show that the data are in good agreement with a χ^2 fitting-value of 1.23 (Supplementary Figure S5H)." To better appreciate the quality of the agreement between the scattering data and the fitting curve, please provide the fitting residuals.

Fig S5 panel H has been edited to include the fitting residuals.

11) I. 307-309: "indicating that these charged patches of Pla1 and Red1 C-termini are in close proximity in solution and are involved in electrostatic interactions." Plausible explanation indeed, but please be more cautious with the interpretation here. Mutations could also affect the overall structure and re-arrangements can impact the affinity even though these regions are not directly involved.

We agree that there could also be some allosteric effects. Therefore, we have edited the statement to: "...indicating that these charged patches of Pla1 and Red1 C-termini are possibly in close proximity in solution and might be involved in electrostatic interactions."

12) I. 340: "mass spectrometry" not "mass spectroscopy".

We would like to believe that this was an autocorrect error, thanks for pointing it out, we changed it back to spectrometry.

13) Figure 7 and text I. 484-488: Are the KDs derived from the the k_{on} and k_{off} values measured with BLI compatible/coherent with the one measured with ITC? Please make the calculation and provide comments on the comparison.

The KDs derived from BLI k_{on} and k_{off} rates have been provided in Supplementary table S2. The KDs obtained from BLI are approximately 22-24-fold stronger than those obtained from ITC. We reason that immobilization of Red1 or Iss1 on a surface (as done in BLI) decreases the degree of freedom for these proteins for binding to Pla1 as opposed to ITC where the components are mobile and free in solution. A statement has been added to the discussion. The other possibility could be that in BLI we used the full-length Pla1 for measurements of kinetics (better signal to noise ratio) while in ITC only the Pla1_{RRM} domain was used. However, we have no evidence suggesting that the Pla1 NTD and MD are involved in binding to Red1₂₈₈₋₃₄₅ or Iss1. In addition, we made control experiments using Pla1_{RRM} domain in BLI and obtained similarly high KDs like those with Pla1 full-length protein in BLI experiments although the signal was much lower due to the smaller size of Pla1_{RRM} domain compared to full-length protein.

14) Figure 8: Please explain the meaning of the the "?" and "???" signs on the model. It is not clear to what sentence/idea they are exactly referring to in the figure legend.

We slightly modified the model figure (Figure 8), it has only one "?" at the site of Pab2 loading to the extended poly(A)-tail of CUTs. We included the following sentence in the figure legend: "We suggest (marked with a "?" in the figure) that the non-canonical poly(A)-binding protein, Pab2, might be preferentially loaded on these poly(A)-tail extensions to facilitate exosome mediated degradation of CUTs."

15) I. 685-687: For SAXS data, it is important for the sample to be homogeneous. How did the authors make sure that the formed complex is really a 1:1 complex and that none of the partners are in slight excess, to ensure a flawless SAXS data acquisition?

To exactly address this problem, the Pla1-Red1 complex was assembled with a 1.5-fold excess of Red1 and subjected to size exclusion chromatography. The peak corresponding to the complex was then pooled and concentrated prior to analyses using SAXS in Batch mode. A statement giving details of this has been added to the Methods section.

Additional minor modifications:

In addition to the changes that were requested by the reviewers and addressed point-by-point in our responses, we also updated the median poly(A) tail length for mRNAs and AS RNAs, because we found 2 minor errors in our bioinformatic pipeline. However, these minor changes did not affect any of the stated conclusions.

- Line 419 in the revised manuscript, we corrected the median poly(A) tail length of WT mRNAs to 31 (from 32 in original submission) because the median was calculated accidentally only from the first 600 rows of the mRNA transcript table. We corrected this mistake and the correct median value is 31 in WT, the values in the mutants did not change.
- Line 427 in the revised manuscript - we corrected the median poly(A) tail length of AS RNAs to 46 (from 41) in WT; 33 (from 31) in *red1_{W298A/F305A}* and 40 (from 36) in *red1_{A288-345}* mutants and the corresponding Supplementary Figure S8F, due to a small error in our bioinformatic pipeline that annotated AS RNAs. We corrected this error and this resulted in slight changes in the final numbers.
- We also included a short paragraph in the Discussion (line 631-639) about PAXT interaction with PCR2 complex.

REVIEWERS' COMMENTS

Reviewer #1 (Remarks to the Author):

The authors have revised the manuscript to my satisfaction.

Reviewer #3 (Remarks to the Author):

I am overall satisfied with the revised version and with the fact that the authors have addressed my comments.

I still think that for the ITC data, it would have been beneficial, for ease of comparison, to plot all the ITC raw data with the same scale on the ucal/s axe. I will not insist, and let the editor decide on this specific aspect.

The authors have clearly improved their manuscript during the revision. I recommend it for publication.